# DESIGNING REPLICABLE NETWORKING EXPERIMENTS WITH *TriScale*

Romain Jacob
*ETH Zurich*

Marco Zimmerling
*TU Dresden*

Carlo Alberto Boano
*TU Graz*

Laurent Vanbever
*ETH Zurich*

Lothar Thiele
*ETH Zurich*

## Abstract

When designing their performance evaluations, networking researchers often encounter questions such as: How long should a run be? How many runs to perform? How to account for the variability across multiple runs? What statistical methods should be used to analyze the data? Despite their best intentions, researchers often answer these questions differently, thus impairing the replicability of their evaluations and the confidence in their results.

In this paper, we propose a *concrete methodology* for the design and analysis of performance evaluations. Our approach hierarchically partitions the performance evaluation into three timescales, following the principle of separation of concerns. The idea is to understand, for each timescale, the temporal characteristics of variability sources, and then to apply rigorous statistical methods to derive performance results with *quantifiable confidence* in spite of the inherent variability. We implement this methodology in a software framework called *TriScale*. For each performance metric, *TriScale* computes a variability score that estimates, with a given confidence, how similar the results would be if the evaluation were replicated; in other words, *TriScale quantifies the replicability* of evaluations. We showcase the practicality and usefulness of *TriScale* on four different case studies demonstrating that *TriScale* helps to generalize and strengthen published results.

Improving the standards of replicability in networking is a complex challenge. This paper is an important contribution to this endeavor; it provides networking researchers with a rational and concrete experimental methodology rooted in sound statistical foundations. The first of its kind. [†]

## 1 Introduction

The ability to replicate an experimental result is essential for making a scientifically sound claim. Without replicability[1]—that is, the ability to assess the validity of claims reported by other researchers—any performance evaluation is questionable, at best. In networking, replicability is a well-recognized

---

[†] [Authors' contributions (CRediT statement)](#)

[1] Different terminology is used to refer to different aspects of replicability research [9, 61]. In this paper, we refer to replicability as the ability of different researchers to follow the steps described in published work, collect new data using the same tools, and eventually obtain the same results, within the margins of experimental error. This is usually called replicability [1] but sometimes referred to as reproducibility.

Table 1: A non-exhaustive list of factors hindering replicability, and selected networking references addressing them.

| | |
|---|---|
| *Focus of this paper* | |
| Variability in experiment design and data analysis | [6, 59, 65] |
| *Other factors hindering replicability* | |
| Lack of documentation | [6, 44, 76] |
| Artifacts unavailability/unusability | [23, 29] |
| Uncontrollability of the exp. conditions | [18, 53, 57, 86, 87] |
| Variability of hardware and software behavior | [12, 50, 75] |

problem which stems from several factors (see Table 1).

To be replicable, performance evaluations must account for the inherent variability of the experimental conditions—*i.e.*, the environment in which the experiment takes place—and the variability in hardware and software behavior in the system under test as well as in the measuring system. To facilitate this, the networking community has put great efforts into developing testbeds and data collection frameworks, *e.g.*, [57, 86, 87]. In addition, several calls for actions have been made to foster proper documentation [6, 65] and artifact sharing [1, 23] which are both essential for replicability.

A more subtle but nonetheless important hindering factor for replicability are *differences in the methodology* used to design an experiment, analyze the resulting data, and draw conclusions from the evaluation. The literature related to this problem is currently limited to generic guidelines [6, 54, 65] and recommendations [39, 44, 59], which leave open several critical questions *before* (How many runs? How long should a run be?) and *after* experiments are conducted (How to process the data and analyze the results?). Without a concrete methodology, networking researchers often design and analyze similar experiments in different ways, making them hardly comparable [13]. Yet, strong claims are being made ("our system improves latency by $3\times$") while confidence is often discussed only in qualitative ways ("with high confidence"), if at all [75, 86]. Furthermore, it is unclear how to effectively assess whether an experiment is indeed replicable. We argue that a concrete methodology is needed to help resolve this situation.

Hence, we developed such a methodology for the design and analysis of performance evaluations for networking research. This paper presents *TriScale*, an implementation of our methodology into a software framework making the

methodology readily applicable by researchers. As further discussed in § 7, we do not claim that our methodology is fitting all situations, nor that it is the best one possible; however, we do find it useful in many practical cases (§ 5). At a high level, our methodology has four key desirable properties.

**Rationality** The methodology rationalizes the experiment design by linking the design questions (*e.g.*, How many runs?) with the confidence in the performance claims.

**Robustness** The methodology is robust against the variability of the experimental conditions. The data analysis uses statistics that are compatible with the nature of networking data and are able to quantify the expected performance variation shall the evaluation be replicated.

**Generality** The methodology is applicable to a wide range of performance metrics, evaluation scenarios (emulator, testbed, in the wild), and network types (wired, wireless).

**Conciseness** The methodology describes the experimental design and the data analysis in a concise and unambiguous way to foster replicability while minimizing the use of highly treasured space in scientific papers.

*TriScale*'s methodology is based on an analysis of the temporal characteristics of variability in networking experiments, which we argue can be decomposed into three timescales. For each timescale, *TriScale* applies a set of appropriate and rigorous statistical methods to derive performance results with *quantifiable confidence*. For each performance metric, *TriScale* computes a variability score that estimates, with a given confidence, how similar the results would be if the evaluation were replicated. The rest of this paper presents *TriScale* and its underlying methodology. More specifically:

§ 2 motivates the importance of the methodology in experiment design and data analysis using a concrete example from the literature, before introducing *TriScale*.

§ 3 covers important statistics basics for replicability.

§ 4 details the inner workings of *TriScale*.

§ 5 illustrates the practicality and usefulness of *TriScale* with four different case studies using testbed experiments or network emulations: congestion control, wireless embedded systems, failure detection, and video streaming.

§ 6 provides concrete recommendations to select *TriScale*'s parameters; that is, how to use *TriScale* in practice.

§ 7 discusses the limitations and possible future developments of the methodology presented in this paper.

With *TriScale*, we provide a concrete methodology that *concretely guides* networking researchers through the design of their experiments and the analysis of the gathered data, while *quantifying the replicability* of the performance evaluation. Hence, *TriScale* complements prior work toward replicable networking research that mostly focused on data collection, *e.g.*, [57, 86, 87].

**Reproduce me!** We strive to make this paper fully reproducible. All data and source code are openly available [36, 37]. Most plots were created using *TriScale* and are *interactive*: the plots themselves are hyperlinks to online versions allowing for dynamic data visualization.

## 2 Overview of *TriScale*

This section first illustrates how *TriScale* improves the analysis of experimental results with a concrete example (§ 2.1) then presents the core principles of the methodology (§ 2.2).

### 2.1 How *TriScale* Improves Data Analysis

Assume you are new to the field of congestion control and would like to understand the strengths and weaknesses of the state of the art. Luckily, the community has developed useful tools like Pantheon [87], a data collection framework that facilitates comparisons of congestion-control schemes.

You are particularly interested in the throughput and one-way delay of full-throttle flows, *i.e.*, flows whose performance is only limited by the congestion control. You start with one flow and evaluate performance using MahiMahi [55], a traffic and network emulator integrated in Pantheon, using the same settings as in [87]: 10 runs of 30 seconds each for all the congestion-control schemes available. Pantheon assists you in collecting the data, but not in their analysis or interpretation. Yet, these are two non-trivial tasks. For example, consider the results shown in Fig. 1a (replicated from [87]) where the dots indicate the mean performance across all runs for two metrics: the mean throughput and 95th percentile of the one-way delay; the ellipses show the 1σ variation across runs, where σ is the standard deviation. Multiple questions arise:

(Q1) Can the schemes be compared? It appears that *TCP Vegas* performs better than, *e.g.*, *TaoVA-100x*. However, since the ellipses capture the results' variability, what can we conclude about the actual performance of these schemes? Can we conclude anything when the ellipses are overlapping? *E.g.*, can we say that *TCP Vegas* performs better than *PCC-Expr*?

(Q2) What is the confidence in the comparison? Intuitively, the results of, *e.g.*, *PCC-Allegro*, which have a large variability, are less trustworthy than those of, *e.g.*, *FillP-Sheep*, for which the ellipse is hardly visible. How does the difference in variability affect your confidence in the overall comparison? Can you quantify this confidence?

(Q3) Is a runtime of 30 seconds sufficiently long to fairly compare the different schemes?

These questions relate to the robustness and rationality challenges (§ 1) and are left unanswered by the analysis shown

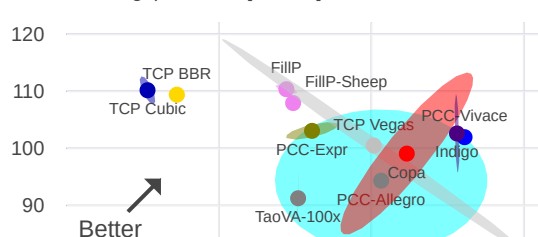

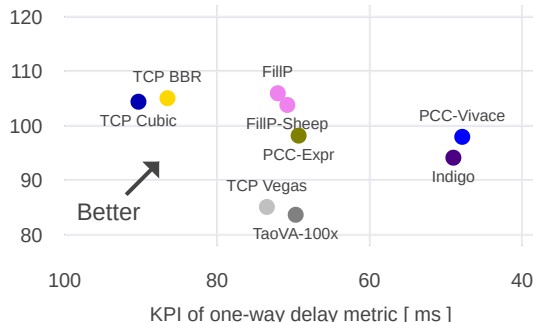

(a) Data analysis with Pantheon (replicated from [87]). *Dots represent the mean performance across all runs; metrics are the mean throughput and 95th percentile of the one-way delay; ellipses represent the 1σ performance variation across all runs, where σ denotes the standard deviation.*

(b) Data analysis with *TriScale*. *Dots represent Key Performance Indicators (KPIs) across all runs: the 25th percentile of the throughput metric and the 75th percentile of the one-way delay metric (same metrics as in Fig. 1a). KPIs are estimated with 75% confidence.*
Online: tiny.cc/triscale-plots#Figure-1

Figure 1: Sample data from our congestion-control case study (§ 5.1). *The same data may be analyzed in different ways. Compared with Pantheon's analysis (Fig. 1a), TriScale's analysis allows for a more intuitive interpretation of the results (Fig. 1b). The performance of each scheme is reduced to a single point, a TriScale's KPI, which simplifies the comparison between the schemes. These KPIs are not arbitrary; they are robust non-parametric statistics estimating, with a given confidence level, the expected performance if the experiment was repeated. Thus, TriScale's KPIs inherently account for the variability in the results.*

in Fig. 1a. In fact, the figure may even suggest wrong interpretations. Ellipses are two-dimensional representations of the standard deviation across runs, which one may interpret that about 68% of the data points to fall in that region. This is wrong for two reasons: first, in 2D, the correct value is $\sim 40\%$;[2] second, this would be correct *only if* the underlying distributions are normal, which is hardly ever true (§ 3).

Fig. 1b illustrates the *same data* analyzed with *TriScale*. The dots now represent *TriScale*'s key performance indicators (KPIs). A KPI estimates a given percentile of a performance metric's underlying distribution—*i.e.*, the unknown distribution we would obtain with infinitely many samples—with a certain confidence. We use the same performance metrics: the mean throughput and the 95th percentile of the one-way delay, for which we have 10 samples (one per run). Based on these 10 samples, instead of computing the mean and standard deviation, *TriScale* computes two KPIs: the 25th percentile of the throughput metric (*higher* throughput is better) and the 75th percentile of the one-way delay metric (*lower* delay is better), which we aim to estimate with a 75% confidence level.[3] In other words, with a 75% confidence, 75% of the runs yield a

performance that is at least as good as the KPI values (*e.g.*, equal or higher throughput). Hence, we use the same performance metrics (mean throughput and 95th percentile of the one-way delay), but a different aggregation strategy (KPIs instead of mean and standard deviation). Note that, in this paper, we simply consider multiple performance dimensions (*e.g.*, throughput and delay) independently. The approach can be extended towards multi-objective performance evaluations using the principles of Pareto-dominance, but such extension is beyond the scope of this paper.

Using the methodology presented in this paper, *TriScale* allows to answer the three questions mentioned previously:

(A1) Since the KPIs are individual dots, we can unambiguously compare the different schemes. Differently to what Fig. 1a suggests, we observe in Fig. 1b that *TCP Vegas* is not strictly better than *TaoVA-100x*: *TCP Vegas* performs worse in terms of one-way delay; also, *PCC-Expr* performs better than *TCP Vegas* in both performance metrics.

(A2) The confidence level of the KPIs explicitly state how confident we are with these results. The independence of measurements is empirically tested, which guarantees the soundness of the performance estimation; data from, *e.g.*, *Copa* appears correlated and are therefore not shown in Fig. 1b.

(A3) *TriScale* tests whether the different schemes have converged (§ 4.5), *i.e.*, the metrics have reached stable values within the experiment; 30 s are actually not enough for certain schemes (§ 5.1), which biases the comparison.

---

[2]In the general case, *i.e.*, multivariate with arbitrary covariance matrix, the natural generalization of the "normalized distance from the mean" is given by the Mahalanobis distance [84]. It follows that the generic "*m*-sigma rules" of *n*-dimensional hyper-ellipsoids can be computed as $Q(n/2, 0, m^2/2)$ where $Q$ is the generalized regularized incomplete gamma function.

[3]75% is a rather low confidence value (95% would be more common). However, estimating the 25th and 75th percentiles with 95% confidence requires at least 11 data points (see Eq. (4)) whereas Pantheon performs series of 10 runs. To compare *TriScale*'s and Pantheon's analysis methods in this example, we chose to lower the confidence level and keep the same number of samples.

**Summary.** Tools like Pantheon [87] support data collection, but leave up to the researcher key parts of the experiment design (How many runs to perform?) and data analysis (How to synthesize results?), leading to ambiguous interpretations and non-replicable results. To address this problem, this paper presents a concrete methodology implemented in *TriScale*.

## 2.2 Core Principles of *TriScale*

*TriScale* is a framework for networking experiments (Fig. 2); it is based on a methodology that streamlines the design and analysis of performance evaluations to improve the replicability of networking evaluations. Its hierarchical approach partitions the performance evaluation in a sequence of stages that build on top of each other and follow the principle of separation of concerns [85]. Specifically, it splits a performance evaluation into three timescales, hence the name **Tri***Scale*.

Given the user's objectives (*e.g.*, the KPIs to analyze and the confidence levels to reach), *TriScale* helps answer questions such as: How many runs should be done? How long should the runs be? When to perform the runs? Based on the answers, the user can then proceed with the data collection. In the analysis phase, the user provides those data to *TriScale* which automatically produces expressive and easy-to-interpret performance reports together with variability scores that quantify the replicability of the evaluation.

In the rest of this section, we explain *TriScale*'s main building blocks. We start by describing the three timescales underlying the methodology, then describe how *TriScale* concretely supports the users with the design and analysis of their performance evaluations.

**Timescales.** We structure *TriScale*'s methodology around three timescales: *runs*, *series* (of runs), and *sequels* (of series). These timescales intuitively capture the different sources of variability underlying performance evaluations in networking.

A *run* is one execution of an evaluation scenario, *e.g.*, a 30s execution of *TCP BBR*. During a run, some performance dimensions are measured, *e.g.*, packet delay, which vary due to different sources of variability such as protocol dynamics and cross-traffic. The performance during a run is summarized by a *metric*, for example, the 95th percentile of the run's measurements. Depending on the scenario, one may want the metric to estimate long-term performance, for example, in case of long-lasting flows; the run should then be sufficiently long to let the metric value converge.

Typically, one executes multiple runs to measure performance; we call such a set of runs a *series*. For example, one may execute 100 runs within one week, from which one obtains a set of metric values, one for each run. We summarize the performance of a series with a *key performance indicator (KPI)* that measures the expected performance for any run by estimating, *e.g.*, the median of the metric distribution within the time span of the series (*i.e.*, the time interval in which runs are performed; *e.g.*, one week). The intuition is that

with a series of runs one randomly samples the distribution of possible experimental conditions during that week, which allows to estimate the distribution of a performance metric.

In general, variability sources such as cross-traffic vary with an a priori unknown temporal long-term correlation; *i.e.*, the distribution of conditions during a series may not be stationary but time-varying. Therefore, in order to generalize the results, one should perform multiple series, which we call *sequels*. Intuitively, sequels allow to estimate the expected performance for any series (*e.g.*, the expected KPI for any week). Our method uses sequels to compute a *variability score* that serves to *quantify the replicability* of an experiment by computing a confidence interval for the expected results one would obtain shall new series of runs be performed.

*TriScale* uses these three timescales of runs, series, and sequels to structure the experiment design and data analysis.

**Experiment design.** The design phase starts with the definition of the evaluation objectives (Fig. 2, left). For each performance dimension, the user defines the metric, the convergence requirements, a KPI, and a variability score (§ 4). Given these inputs, *TriScale* derives the minimum number of runs (*#runs*) and series (*#series*) needed to compute the chosen KPIs and variability scores, thus answering the question of how many runs to perform. Using data from test runs or previous experiments, *TriScale* can assess whether the runtime appears long enough to let the metric values converge. Additionally, *TriScale* can make use of these test runs to identify time-dependent patterns in the experimental conditions (§ 4.6). This is important in order to understand the root cause of the statistical behavior of the measurements, and helps to answer the question of *when* the runs should be performed. Note that the congestion-control example presented previously uses network emulation; thus, there is no time dependency, and it does not matter when the experiment is performed (*i.e.*, *span: anytime*). The design phase produces in a report (Fig. 2, right) summarizing how to run the experiments. Based on this report, the user can collect the raw data and then moves on to the analysis phase.

**Data analysis.** Once the experiment has been designed and the data collected, the raw data are passed to *TriScale* for a three-stage analysis, one per timescale. First, the raw data from one run are processed, *i.e.*, convergence is assessed, and the performance metrics are computed, producing one number per run and per metric. The short-term variability in the experimental conditions is accounted for by performing a series of runs. This timescale leads to one number per series and per metric: the KPIs (§ 4.2). Finally, the sequels (repetition of series) are used to compute a *variability score* capturing the long-term variability of the KPIs. This timescale leads to one number per metric (§ 4.3).

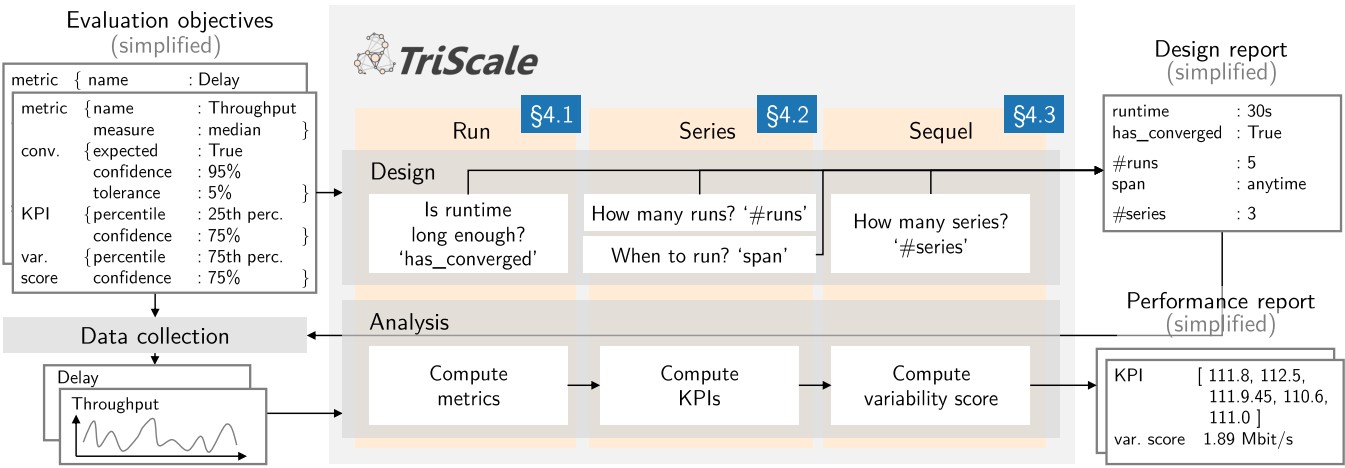

Figure 2: Overview of *TriScale*. *TriScale is a framework supporting the design and analysis of networking experiments. TriScale assists the user in the design phase with a concrete methodology to answer important experiment design questions such as "How many runs?" and "How long should the runs be?" After the raw data are collected, TriScale supports the user by automating the data analysis. The framework implements robust statistics that handle the intrinsic variability of experimental networking data and returns expressive performance reports along with a variability score that quantifies the replicability of an experiment.*

**Readily usable!** *TriScale* is implemented as a Python package [36]. For each timescale, a dedicated function takes raw data as input, performs the corresponding test or analysis, returns the results, and produces optional data visualizations such as those shown in Fig. 3 to 5. We aimed to make *TriScale* intuitive and easy to use. For a better impression of its usability, you can run an interactive demo directly in your web browser [38]. See Appendix § A for more details.

## 3 Statistics for Replicability

This section briefly reviews classes of statistical approaches and motivates the choice of the methods we use in *TriScale* to handle the variability inherent to networking evaluations. Note that, in this paper, we consider a safe evaluation approach: we do not suppose any knowledge about the statistical distributions underlying the variability of the measurements. Of course, tighter estimates are possible if additional reliable information is available, which would lead to a different instance of the framework (discussed in § 7).

**Descriptive and predictive statistics.** A statistic is a number computed from some data using a mathematical formula; it can always be calculated and provides a factual description of the underlying data. This is referred to as a *descriptive statistic*. In addition, certain statistics have some *inference* power; *i.e.*, based on the collected data, one may infer the shape of the (unknown) underlying data distribution. These are then referred to as *predictive statistics*.

Predictions are always uncertain and rely on specific hypotheses. If the hypotheses hold for the collected data, then predictive statistics estimate, with a quantifiable level of confidence, some property of the underlying distribution such as the mean or the median. One can then predict expected values of data samples that have not been collected. A common hypothesis is that the collected data is *independent and identically distributed* (*i.i.d.*). Informally, this means that the underlying distribution of the data does not change and that successive data samples are uncorrelated (past samples do not say anything about future samples). It is also common to presume the *nature of the distribution*, *e.g.*, a normal distribution. For example, one can estimate the mean $\mu$ and standard deviation $\sigma$ of a distribution based on an *i.i.d.* data sample. If the underlying data distribution is normal (the hypothesis), we can infer that about 68% of all data points will be contained within $\mu \pm \sigma$ (the prediction). However, if the distribution is *not normal*, the statistics $\mu$ and $\sigma$ are *only* descriptive—they do not predict anything about unseen samples.

**Statistical methods.** There are two common classes of statistical approaches: hypothesis testing and estimation.

*Hypothesis testing* consists of formulating a so-called null hypothesis that the test aims to reject. Based on the collected data, one computes the probability, called the *p*-value, that the null hypothesis is correct. If the *p*-value is sufficiently low, the null hypothesis is rejected and considered proven incorrect. *E.g.*., the one-way ANOVA [81] is a common method to test for significant differences in the mean of multiple samples.

*Estimation* consists of computing confidence intervals (CIs) for a given parameter (*e.g.*, the mean of a distribution). A CI is always associated with a certain confidence level (*e.g.*, a

95% CI) which can be seen as the probability that the interval includes the true value of the parameter; *e.g.*, $[a, b]$ is a 95% CI for the mean if the true mean value is between $a$ and $b$ with a probability of at least 95%.[4]

These approaches are further classified as *parametric* if the nature of the underlying distribution is known and as *non-parametric* if no assumptions are made about the underlying distribution. For example, the Kruskal-Wallis test [80] is the non-parametric equivalent of the one-way ANOVA. The tests are similar, but the former does not assume that the underlying distribution is normal. The central limit theorem [83] offers another alternative to handle unknown distributions, but it only allows to argue about the arithmetic mean.

**Statistics for replicability in networking.** Informally, replicability is the principle that the "same experiment" leads to the "same results." Thus, assessing replicability entails predicting whether future data (the results of a newly-performed experiment) will be the same as the known data (the results of previous experiments); it is a prediction. One important idea of *TriScale* is to try to predict the expected amount of variability in an evaluation, and to use this prediction as a measure of replicability.

Literature reports that experimental data are rarely normally distributed and hence recommends using *non-parametric* statistics [50, 66]. One should also consider *robust statistics* (*e.g.*, using median instead of mean), *i.e.*, statistics that are not overly skewed by outliers, which are common in experimental networking data. While hypothesis testing is common, statisticians argue that the methods is misunderstood and misused [45] and are thus calling for a change in practices [24, 78]. We favor *estimation* over testing because CIs are more legible than *p*-values and easier to interpret. Furthermore, the confidence level of an estimation only depends on the sample size, which is useful to guide the experimental design.

In 1936, Thompson introduced a method to compute non-parametric CIs for percentiles [74]. This approach is found in statistics [25] and computer science [46] textbooks, but it is rarely used today ( [10, 50, 66] are the few exceptions). As Thompson's method is well-suited to handle the variability of experimental networking data, we use it in the described instance of *TriScale*'s methodology (§ 4.5). We illustrate the potential of the approach (§ 5) and facilitate its use by providing the necessary software support (§ A).

---

[4]Note that this is a frequentist probability: that is, for many repetitions of the distribution sampling, if $[x_n, x_m]$ is a 95% CI for the mean (with $n$ and $m$ two sample indices), then the true mean value will be contained in $[x_n, x_m]$ approximately 95% of the time. However, once a specific sample is collected, it is no longer mathematically correct to talk about probability: the distribution mean has an exact—albeit unknown—value which is thus contained in a given numerical interval $[a, b]$ with "probability" of either 0 or 1. This is not an issue per se, but simply a semantic clarification: a confidence level is not *exactly* a probability, although the two are often confounded.

## 4 Designing *TriScale*

In this section, we first describe the data analysis performed by *TriScale* and how the analysis procedure is linked to the design of an experiment (§ 4.1 to § 4.3). We then illustrate how the formalism introduced by *TriScale* allows to unambiguously describe an entire performance evaluation with only a handful of parameters (§ 4.4). We further detail the robust and non-parametric statistical methods used by this instance of *TriScale* (§ 4.5), and discuss how the framework assists a user in deciding the required time span for a series of runs (§ 4.6). Finally, we discuss how *TriScale*'s variability score allows to assess the replicability of experiments (§ 4.7).

### 4.1 Runs and Metrics

In *TriScale*, metrics evaluate a performance dimension across a run; for example, the mean throughput achieved by a congestion-control scheme over 30 s runtime of a full-throttle flow. Computing a metric takes the following inputs.

**Inputs.**
- The metric *measure*, *e.g.*, mean, maximum;
- The *convergence* requirements
  {    expected : True/False ,
      confidence : $C$ (*default: 95%*) ,
      tolerance : $t$ (*default: 5%*)    };
- The raw data of the run.

In general, any measure can be used. The current implementation of *TriScale* (§ A) supports the arithmetic mean, the minimum, the maximum, and any percentile. The definition and usage of the confidence and tolerance are detailed with the convergence test (§ 4.5).

**Procedure.** If the run is expected to converge,[5] *TriScale* starts by performing a convergence test (§ 4.5) whose purpose is to estimate whether the metric has reached a stable value by the end of the run—and if it is thus a reliable estimate of the long-running performance. Note that the performance dimensions and convergence behavior can vary between systems. Therefore, suitable methods to test for convergence may vary and need to be considered during the design of an experiment. The approach presented in this paper does appear well-suited to a variety of networking experiments (§ 5).

The implemented convergence test starts by computing metric values over a sliding window of the raw data points, with a fixed size of half the data points. For each window, one metric value is computed, starting with the first half of the data. The window repeatedly slides by a 100th of the number of points until all data are used, leading to a set of 100 metric values. *TriScale* performs its convergence test (detailed in § 4.5) on these metric values. Note that this procedure tests

---

[5]Not all runs should necessarily converge. For example, consider the evaluation of an FTP client by downloading a 10MB file. One may be interested in the throughput during the file transfer (*e.g.*, to study fairness), but it does not matter whether the throughput actually converges, since there is a finite task to perform.

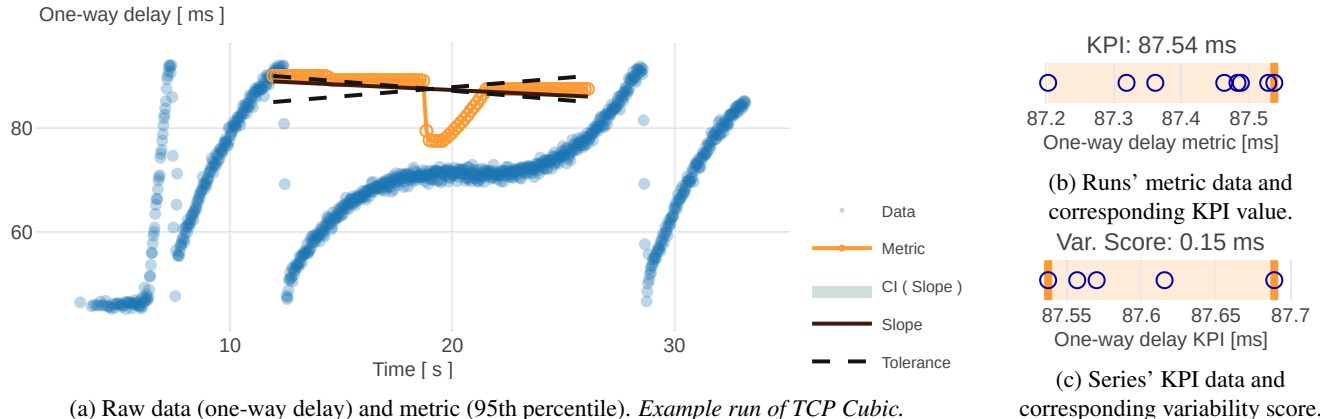

(a) Raw data (one-way delay) and metric (95th percentile). *Example run of TCP Cubic.*

(b) Runs' metric data and corresponding KPI value.

(c) Series' KPI data and corresponding variability score.

Figure 3: Example plots produced by *TriScale* during the data analysis. *Fig. 3a: computation of the metric (95th percentile on one-way delay) with convergence test (confidence 95%, tolerance 5%). Fig. 3b: computation of the KPI (75th percentile with 75% confidence). Fig. 3c: computation of the variability score (25-75th percentile range with 75% confidence). Sample data from the case study in § 5 (TCP Cubic).* Online: tiny.cc/triscale-plots#Figure-3

the convergence of the *metric*—which is the focus of the analysis—and not of the raw data. Using a sliding-window approach helps reduce the impact of the transient behavior in the raw data on the convergence test. If the test is passed, *TriScale* returns the median of the converged metric values as run metric. If convergence is not expected, *TriScale* simply computes the run metric over the whole raw data.

**Outputs.**
- The result of the convergence test (if any);
- The metric value for the run;
- Textual logs, plot of the input and metric data.

**Link to the experiment design.** The computation of metrics is linked to the definition of the *runtime*, *i.e.*, how long a run should be. If the evaluation scenario is finite (*e.g.*, transmit 1 MB of data), the runtime must be long enough to complete the task. If the evaluation is long-running (*e.g.*, estimate battery lifetime), the runtime must be long enough for the metric (*e.g.*, energy consumption) to converge. Details about the specific convergence test are described in § 4.5. As illustrated in § 5, *TriScale* can analyze experiments to estimate whether the runtime appears long enough *i.e.*, it can assess with quantifiable confidence that the metric values are stable for a given runtime. However, *TriScale cannot guarantee* that the runtime is long enough for a sound evaluation of long-running performance, as this requires context-specific knowledge.[6]

## 4.2  Series and KPIs

*TriScale*'s KPIs evaluate performance across a series of runs. Performing multiple runs allows to mitigate the inherent variability of the experimental conditions. KPIs capture this variability by estimating percentiles of the unknown metric distributions. Concretely, in *TriScale*, a KPI is a one-sided CI of a percentile; *e.g.*, a lower bound for the 75th percentile of the throughput metric estimated with a 95% confidence level.

**Inputs.**
- The KPI definition
  { percentile : $p$ ,
    confidence : $C$  };
- The metric values from a series of runs.

**Procedure.** To compute a KPI (*i.e.*, to compute a CI for a given percentile), *TriScale* uses Thompson's method (§ 4.5) which requires the input data to be *i.i.d.* Therefore, *TriScale* starts by performing an independence test (§ 4.5) to check that the metric data do appear empirically *i.i.d.*

**Outputs.**
- The result of the independence test;
- The KPI value for the series of runs;
- Textual logs, plot of the metric and KPI data.

**Link to the experiment design.** The computation of KPIs is linked to the definition of the number of runs in a series (*# runs*) and the series time span (*span*). The minimal number of runs in a series directly follows from the definition of the KPI, *i.e.*, the percentile to estimate $p$ and the desired confidence level $C$ (see Eq. (4)). The series time span refers to the time interval used for scheduling the runs in a series; *i.e.*, when to run the experiment. This is important because networks often feature time-dependent conditions; for example, there may be systematically more cross-traffic during daytime than nighttime. Failing to consider such dependencies may bias the results and yield wrong conclusions. This concept of series also applies when "slicing" a long experiment into smaller independent ones. In such a case, it is crucial to consider warm-up and cool-down effects to avoid biasing the results. Note that such slicing strategy is more likely to result in empirically non-*i.i.d.* data than a random schedule of truly independent runs. *TriScale* helps to detect certain classes of

---

[6]For example, if a system is configured to switch from its bootstrapping to its steady-state behavior after *e.g.*, an hour, and if we test for only a few minutes, it is impossible for *TriScale* to "predict" the behavior change; it is limited to what is observed.

dependencies with a dedicated "network profiling" function (example in § 5). Here, again, other dependency analysis methods can be implemented to tailor *TriScale* to a specific class of systems under evaluation.

## 4.3 Sequels and Variability Score

*Sequels* are repetitions of series of runs. *TriScale*'s variability score evaluates the variations of KPI values across sequels. Sequels enable *TriScale* to detect long-term variations of KPIs and ultimately to quantify the replicability of an experiment.

Concretely, a variability score is made of two one-sided CI for a symmetric pair of percentiles; *e.g.*, a 75% confidence interval for the 25-75th percentile range of the delay KPIs from all sequels. Again, we attach a confidence value to the CI or, equivalently, to the percentile estimation.

**Inputs.** • The variability score definition
{  percentile : $p$  (or 1-$p$),
   confidence : $C$            };
• The KPI values of each sequel.

**Procedure.** The procedure is the same as for the KPIs: *TriScale* first performs an independence test on the KPI data before computing the variability score.

**Outputs.** • The result of the independence test;
• The variability score value across all sequels;
• Textual logs, plot of KPI values, and corresponding variability score.

**Link to the experiment design.** The computation of the variability score is linked to the definition of the number of series (#*series*). The minimal number of series directly follows from the definition of the variability score; *i.e.*, the percentile to estimate $p$ and the desired confidence level $C$ (Eq. (4)).

## 4.4 Formalism Brings Conciseness

*TriScale* formalizes the definition of the evaluation objectives. As illustrated in Fig. 2, for each performance dimension, the user defines a metric together with its convergence requirements, a KPI, and a variability score. *TriScale* links these objectives with the experiment design, resulting in four additional parameters: the number of runs per series (#*runs*), the number of series (#*series*), the length of a run (*runtime*), and the time span of a series (*span*).

With this formalism, *TriScale* provides *conciseness*: 12 parameters are sufficient to formally describe the entire performance evaluation. Since the data analysis in *TriScale* is automated and deterministic, documenting these parameters guarantees computational reproducibility, *i.e.*, the ability to recreate the results when all raw data are available [48].

Table 2 shows a few examples of concrete parameter settings for typical networking evaluation use cases. For example, evaluating the latency of a real-time protocol requires high confidence levels for extreme percentiles. This quickly

**Why not just one big series?** A common practice today is to perform *one* series of many runs (say 100). The problem with this approach is that it does not allow estimating replicability, *i.e.*, what the expected performance is shall one re-do the experiment (*i.e.*, one series of 100 runs). Sequels are meant to address this problem: by running several *independent* series (*e.g.*, 10 series of 10 runs),[a] one can estimate how much the performance varies across series and thus assess replicability (§ 4.7). This, of course, comes at a cost. If the total number of runs remains fixed (*e.g.*, 100), the KPI estimates for each series will be worse, *i.e.*, resulting in wider CIs and/or using lower confidence levels—there is no free lunch.

---

[a]The statistical analysis requires the KPI values to be *i.i.d.* Therefore one should *not* perform one batch of 100 runs and simply split them into chunks of 10 runs to produce 10 series, as this is likely to induce correlation between the series. The same holds true for "making up" multiple runs by slicing a large measurement, *e.g.*, making 60 1-minute runs out of a measurement of one hour. To hold statistically relevant information, runs and series must be collected independently of one another (as much as possible).

increases the number of runs that must be performed, *e.g.*, at least 90 for estimating the 95th percentile with 99% confidence and at least 299 for estimating the 99th percentile with 95% confidence. This illustrates that it is "easier" to increase the confidence level of an estimation than to estimate a more extreme percentile with the same confidence level. Note that both *#runs* and *#series* are only derived from the definition of the KPI and the variability score; *i.e.*, these parameters are not influenced by the runtime or the time span of an experiment.

The second use case in Table 2 (bottom rows) illustrates two different perspectives on "averages" using delay as an example. If one uses the median and the 90th percentile as metric and KPI, respectively, one can conclude that 90% of the runs have a median delay equal or better than the KPI value. Conversely, if one uses the 90th percentile as metric and the median as KPI, one can conclude that, in half of the runs, the 90th percentile of the delays in a run is equal or better than the KPI. Both are "averages," but with different meanings and different requirements in terms of number of runs. Only users can know what is more appropriate for their evaluation, but it is important to understand this distinction when designing it.

## 4.5 Statistics in *TriScale*

As discussed in § 3, performance evaluations in *TriScale* focus on statistics that are both robust (*i.e.*, tolerant to outliers) and non-parametric (*i.e.*, which make no assumption about the nature of the data distribution). If reliable information about the underlying distribution of the data is available, one can use parametric approaches to produce tighter estimates—

Table 2: Exemplary evaluation of typical parameters for networking. *TriScale returns the minimal number of runs (#runs) and series (#series) based on the definition of KPI and variability score, respectively.*

| Use case | Metric | Convergence | | | KPI | | Var.Score | | Experiment Design | | | |
|---|---|---|---|---|---|---|---|---|---|---|---|---|
| | | | | | | | | | **Experiment Design** | | | |
| | Measure | Exp. | Conf. | Tol. | Perc. | Conf. | Perc. | Conf. | #runs* | #series* | runtime | span |
| Latency of | | | | | 95 | 95% | median | 75% | 59 | 3 | | |
| real-time | max | True | 95% | 5% | 95 | 99% | 75 | 75% | 90 | 5 | Depend on | |
| protocol | | | | | 99 | 95% | median | 90% | 299 | 5 | networks and | |
| Average | median | False | - | - | 90 | 95% | median | 90% | 29 | 5 | protocols | |
| delay | 90th perc. | | | | median | 95% | median | 90% | 5 | 5 | | |

see § 7 for more details. However, as this is generally not the case, we focus here on a *TriScale* instance that does not require such information.

The instance of *TriScale* we present in this paper uses three carefully-chosen statistical methods. We first present the convergence test used in the computation of metrics, which is based on the Theil-Sen linear regression [70, 73]. We then introduce the computation of confidence intervals using Thompson's method [74]. Since this method requires the data to be *i.i.d.*, *TriScale* empirically checks whether this requirement is satisfied with an independence test. We conclude with a discussion of the consequences if one of the tests fails.

**Convergence test.** When an evaluation aims to estimate long-running performance—the expected performance if the run would continue for a very long time—one must verify whether the runs appear long enough to produce reliable estimates. To this end, *TriScale* implements a convergence test based on the Theil-Sen linear regression [70, 73]. This approach computes the slope of the regression line as the median of all slopes between any pair of data points. A $C\%$ CI for the slope is defined as the interval containing the middle $C\%$ of slopes. *TriScale*'s convergence test is passed if the $C\%$ CI for the regression is included in the tolerance value ($\pm t\%$). The confidence $C$ and the tolerance $t$ can be specified by the user in the evaluation objectives (see Fig. 2, left) and are otherwise set to 95% and 5% by default, respectively.

Such a test is sensitive to the scale of the input data. To remove this dependency, *TriScale* first maps the data to $[-1, 1]$ using a linear transformation, then performs the convergence test on the scaled data. Hence, the convergence test becomes dimensionless, and the same tolerance value can be used to compare different protocols or systems without bias. Fig. 3a shows an example of the Theil-Sen slope (brown, solid), its CI (light blue, solid), and the tolerance (black, dashed).

Note that this convergence test is based on some assumptions; *e.g.*, that the convergence of metric values is captured by the convergence of the slopes toward zero. This does not hold if one measures, *e.g.*, energy consumption since it is cumulative over time; one should measure power draw instead.

Finally, note that convergence is not *necessary* for replicability; an experiment can be replicable but not "converge," *e.g.*, due to a too short runtime (Fig. 4). However, convergence is required to assess whether the runtime is long enough to produce reliable performance estimates (again, see Fig. 4), which is paramount to fairly evaluate and compare systems.

**Confidence intervals.** *TriScale* defines KPIs and variability scores based on CIs for distribution percentiles, which can be computed using a robust and non-parametric approach based on Thompson's method [74], later shown to be valid for any independent sample of a continuous distribution [25].

Let us denote by $P_p$ the $p$-th percentile of a distribution and by $\mathbb{P}(X)$ the probability of an event $X$. By definition, a data sample $x$ is smaller than $P_p$ with probability $p$ (and larger with probability $1 - p$). For a sorted list of *i.i.d.* samples $x_i$ (where $i = 1...N$), the probability that $P_p$ lies between two consecutive samples follows the binomial distribution [74]:

$$\mathbb{P}(x_k \leq P_p \leq x_{k+1}) = \binom{N}{k} p^k (1-p)^{N-k}, \quad k = 0...N \quad (1)$$

where we assume that $x_0 \to -\infty$ and $x_{N+1} \to +\infty$. From this result, it follows that the probability for $P_p$ to be larger than any sample $x_m$ ($1 \leq m \leq N$) can be computed as:

$$\mathbb{P}(x_m \leq P_p) = 1 - \sum_{k=0}^{m-1} \binom{N}{k} p^k (1-p)^{N-k} \quad (2)$$

Note that these probabilities are symmetric; that is,

$$\mathbb{P}(x_m \leq P_p) = \mathbb{P}(x_{N-m+1} \geq P_{1-p}) \quad (3)$$

Thus, Eq. (2) provides either the upper or lower bound required for computing a one-sided CI. If the probability distribution is discrete, then Eq. (2) becomes an inequality ($\mathbb{P}(x_m \leq P_p) \geq \ldots$ [25]), which provides safe (*i.e.*, conservative) estimates of which sample $x_m$ is the bound of the CI of interest. Furthermore, by plugging $m = 1$ in Eq. (2), one can derive a closed form expression for the minimum number of samples $N$ required for $x_1$ to be a CI (lower bound) for any

percentile $p$ with any confidence level $C$ [66]:

$$N \geq \frac{\log(1-C)}{\log(1-p)} \qquad (4)$$

By Eq. (3), Eq. (4) also gives the number of samples $N$ required for $x_N$ to be a CI (upper bound) for any $p$ and $C$.

*TriScale* leverages Eq. (4) to define the minimum number of runs and series required for estimating the KPIs and the variability scores. This approach provides robust estimates for distribution percentiles and *does not make any assumption on the nature of the underlying data distribution*. It does, however, require that the data samples are *i.i.d.*; thus *TriScale* checks whether this requirement holds with an empirical independence test, described next.

**Independence test.** Estimating the percentile of a distribution often (if not always) requires that the samples are *i.i.d.* This is also the case for Thompson's method [74]. *TriScale* implements an empirical independence test to check whether we can safely treat the samples as *i.i.d.*[7] This independence test is applied to the metric data (resp. KPI data) before the computation of a KPI (resp. a variability score). This poses the particular challenge that the number of data samples may be very small (*e.g.*, 3 or 5 KPI values). *TriScale*'s independence test must therefore not be too strict.

The test proceeds in two steps. First, *TriScale* tests whether the data are *weakly stationary* (*i.e.*, no trend and constant autocorrelation structure [17]). *TriScale* verifies this empirically using its convergence test with a confidence of 50% and a tolerance of 10%; these "loose" parameters are used to compensate for (very) small sample sizes. Second, *TriScale* computes the *sample autocorrelation coefficients*, denoted by $\widehat{\rho}_k$, which measure the linear dependence between values of a weakly stationary data series, where $k$ is the lag between data points. A series of size $N$ is *i.i.d.* with 95% probability if $|\widehat{\rho}_k| \leq 1.95/\sqrt{N}$ for $k \geq 1$ [17].

**What if a test fails?** The user is responsible for designing the evaluation in such a way that the collected data will (likely) pass the tests. *TriScale* facilitates this by guiding the choice of runtime to pass the convergence test and informing about any network time dependencies (§ 4.6) to pass the independence test. Yet, the data may still be correlated or unstable, leading to failing tests (see examples in § 5). Even in such cases, the data may contain useful information. *TriScale*'s metrics, KPIs, and variability scores can be computed. However, since the required hypotheses do not hold, the statistics are *only descriptive* (§ 3); that is, they do not allow to predict the

---

[7]Generally, *i.i.d.*-ness is a property of the experiment design, not of the data. For networking experiments, however, it is often not possible to guarantee independence: *e.g.*, the experimental conditions cannot always be fully controlled and may be correlated. In such cases, it is common to empirically check whether the data are correlated. If the empirical dependence between data samples is sufficiently low, it is considered acceptable to treat the samples as *i.i.d.* [46].

expected performance and, in particular, they cannot—and should not!—be used to assess an experiment's replicability.

## 4.6 Network Profiling

*TriScale* can assist the user in deciding on the time span for a series of runs, *i.e.*, the time interval containing all the runs of one series. This is important in order to avoid biasing the evaluation results with time dependencies in the experimental conditions. Indeed, it is common for networks to exhibit periodic patterns. For example, there may be more cross-traffic (*i.e.*, interference) at specific times of the day. In the statistics literature, these patterns are called *seasonal components*. Neglecting these may bias experiments and lead to wrong conclusions, as illustrated *e.g.*, in § 5.2 and [75].

To address this, *TriScale*'s network profiling function analyzes "network condition data." Informally, such data should be measurements of metrics that capture the "friendliness" of the experimental environment for the system we evaluate. For example, this could be noise floor data (in a wireless testbed) or congestion levels (in a wired network). It is important that these data are collected prior to the performance evaluation and at regular intervals; this may be a significant overhead, but it is necessary to identify possible seasonal components in the experimental conditions. Some academic testbeds regularly collect and make such data available, *e.g.*, [40]. Practically, *TriScale* computes the autocorrelation coefficients of the network condition data. Peaks in the autocorrelation plot suggest seasonal components in the network conditions (see Fig. 5), which helps detect sometimes-unexpected time dependencies.

To avoid biasing the results, the span of a series of runs should be chosen as a multiple of the—assumed, known or observed—seasonal components. The same care must be taken when choosing the time of a run within a series; it is recommended to randomly sample the entire span of a series.

While this profiling approach is designed to identify the dynamics of one environment, it may also be used to compare different ones. For example, a cellular network in a business district and a residential area may exhibit different seasonal patterns. When attempting to replicate an experiment in a different environment, one must adapt the experiment design to account for the dynamics of that specific environment.

## 4.7 Assessing Replicability

Replicability refers to the ability of obtaining "the same" results when performing "the same" experiment. In statistics, such property can be investigated using *equivalence testing* [45], which checks whether the values of some parameter of interest, for example the median, obtained for different samples are sufficiently close to be considered "the same." Unfortunately, there is no general way to define "the same" or even "sufficiently close." One must specify in advance a threshold for the equivalence test based on expertise.

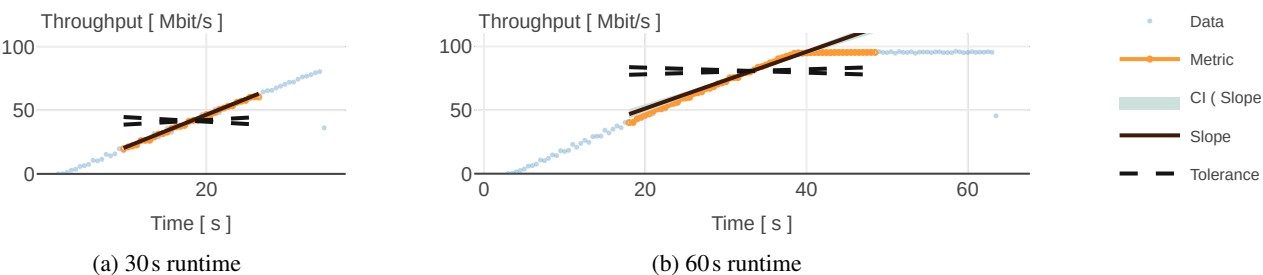

(a) 30 s runtime             (b) 60 s runtime

Figure 4: Egress throughput of the *LEDBAT* congestion-control scheme in MahiMahi [55]. *A runtime of 30 s is clearly not sufficient for LEDBAT's throughput to converge (Fig. 4a). The scheme does converge eventually (Fig. 4b), but even with 60 s runtime, TriScale's convergence test fails as the impact of the start-up phase is too important when all data are considered. Two possible solutions would be to either (i) increase the runtime or (ii) prune the start-up time from the raw data. See Appendix § B.1 for further details.* Online: tiny.cc/triscale-plots#Figure-4

Then, how to assess replicability of networking experiments? How to design a "replicability test" that fairly adapts to different networking contexts and metrics? Setting fixed absolute threshold values does not make much sense. The next natural idea is to consider relative thresholds, *e.g.*, ±5% of the median value. One problem with this approach is that it measures how *stable* the results are, which does not exactly capture the notion of replicability: one system can have large performance fluctuations, but these fluctuations may be stable over time (*e.g.*, the saw-tooth behavior of a congestion window). The performance evaluation of such a system should be assessed as replicable, but a relative threshold would be biased to rule against it. Moreover, setting appropriate values (*e.g.*, 5%) appears difficult—if not impossible—to do in a context-agnostic manner and would work against our objective of generality. We conclude that defining a generic threshold for equivalence testing in networking might not be possible. But it may also be unnecessary!

We argue that it is more important to confidently estimate the variability of the results, which *TriScale* computes with its variability score (§ 4.3). This score *quantifies replicability*: the larger the score, the less replicable are the results (see the example in § B.1). Shall a binary cut between "replicable" and "not replicable" be desired, a threshold can be set based on the variability score, *e.g.*, "Results are said replicable when the variability score is less than 20 Mbps." Clearly, such a threshold can only be context-specific. Thus, deciding on threshold values is more related to benchmarking and therefore goes beyond the scope of *TriScale* (see § 7).

## 5 *TriScale* in Action

We present four case studies which illustrate shortcomings in performance evaluations that *TriScale* addresses (§ 5.1–5.2), and show how *TriScale* allows generalizing performance claims with a quantifiable confidence (§ 5.3–5.4). Further details on these case studies (*e.g.*, link to datasets, plots) are available in Appendix § B. Finally, we show that *TriScale*'s data analysis induces no significant overhead (§ 5.5).

## 5.1 Congestion Control

The first case study illustrates that, for estimating long-running performance, it is important to carefully set the length of runs (the runtime) and to check whether the performance has converged for the system under evaluation.

We continue the evaluation introduced in § 2.1, which compares congestion-control schemes using Pantheon [87]. Assume we are now interested in *long-running flows*; that is, our goal is to estimate the performance one would obtain if the flows ran "forever." *TriScale*'s convergence test (§ 4.1) checks whether the length of a run is long-enough to provide a robust estimate. Since all schemes are different, it is hard to know a priori the minimum runtime for which the schemes actually converge. For this reason, we test runtimes from 10 to 60 s and check when the schemes pass the test.

For a runtime of 30 s (used by the maintainers of Pantheon [58]), only 11 out of 17 schemes pass the test (*i.e.*, converge) in most of the cases. *Verus*, *PCC-Allegro*, and *Copa* only converge in less than half of the runs (see § B.1), whereas *QUIC Cubic*, *TCP Vegas*, and *LEDBAT never* pass the test, even with a runtime of 60 s. Fig. 4 details the case of *LEDBAT*. The functioning of this congestion-control scheme causes the throughput to ramp-up in the first 38 s and then converge to about 92 Mbps. Thus, if one uses a runtime of 30 s without checking for convergence, the computed mean throughput is about 40 Mbps, which is a totally wrong estimation of *LEDBAT*'s long-running throughput.

Takeaway 1. **Always check for empirical independence; check for convergence whenever necessary.** *TriScale*'s convergence test checks whether the runtime of an experiment appears sufficiently long to estimate the system's long-running performance. Failing tests inform users about the need to increase the runtime or take other measures (*e.g.*, pruning the start-up phase of raw data) in order to avoid wrong conclusions. Independence should never be assumed and always empirically validated.

## 5.2 Wireless Embedded Systems

This case study shows the importance of carefully choosing the time span for a series of runs. In particular, if there are strong temporal patterns in the experimental conditions, one may derive wrong results in spite of high confidence levels.

We run a simple evaluation of Glossy [28], a low-power wireless protocol based on synchronous transmissions [90]. A key parameter of Glossy is the number of retransmissions $N$. We are interested in investigating the impact of $N$ on the reliability of Glossy, measured as the packet reception ratio (PRR), for which we aim to estimate the median value with a 95% confidence level—our evaluation's KPI. Refer to § B.2 for more details. We collect data using the FlockLab testbed [47], which is located in an office building where we expect more interference during daytime than nighttime. To mitigate this effect, we perform series of 24 runs scheduled randomly within one day, one per value of $N$. Computing the KPI leads to a PRR of 88% and 84% for $N=1$ and $N=2$, respectively. In other words, it appears that two retransmissions instead of one *reduces* reliability.

The experiment led to this (incorrect) conclusion because we (intentionally) neglected a weekly seasonal component revealed by *TriScale*'s network profiling function (Fig. 5): there is more interference on weekdays than on weekends. To account for this dependency, we repeat the experiment but extend the overall span to one week; this leads to KPI of 80% and 88% for $N$=1 and 2 respectively, which matches our expectations on Glossy's reliability.

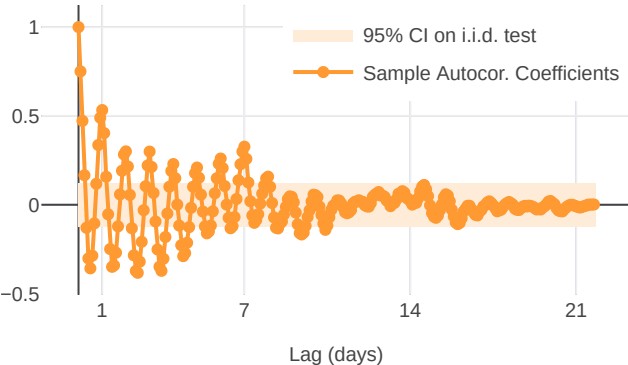

Figure 5: Autocorrelation plot for the wireless link quality on FlockLab [47], based on the raw data collected by the testbed maintainers [40]. *The dataset contains one test every two hours, and we show here the lag in days (i.e., at lag 1, we find the correlation between tests that are 24h apart). The first peak at lag 1 indicates the (expected) daily seasonal component. The data also show another clear peak at lag 7, which corresponds to one week. Indeed, there is less interference in the weekends than on weekdays! See Appendix § B.2 for further details.* Data recording during August 2019. Online: tiny.cc/triscale-plots#Figure-5

Takeaway 2. **Using a high confidence level does not prevent wrong conclusions!** Real networks exhibit short-term variations that are unpredictable and often unavoidable, which is why it is important to perform multiple runs in a series. Moreover, there may also be systematic patterns; *i.e.*, epochs with consistently more or less interference. Knowing about and accounting for these patterns is important to ensure fair comparisons. The time span of a series should be long enough such that it does not matter when the series of runs starts. To avoid biasing the results, the span should be chosen as a multiple of the seasonal components, which can be identified using *TriScale*'s network profiling function.

## 5.3 Failure Detection

This case study illustrates how the methodology of *TriScale* allows generalizing performance claims for large sets of input parameters based on a relatively small sample. We focus on Blink [35], an algorithm that detects failures and reroutes traffic directly in the data plane. The authors evaluated Blink's performance in terms of the true positive rate (TPR—the fraction of failures successfully detected) and the time needed to reroute the traffic based on 15 Internet traces [19, 21] containing data for thousands of prefixes. A subset of prefixes was randomly selected, based on which synthetic traces including artificial failures were generated.

Using *TriScale*, we can generalize the results. For each trace, the evaluation of Blink on one prefix can be seen as a *TriScale* run. Since the prefixes are randomly selected from a fixed set, runs are *i.i.d.*, and we can use *TriScale*'s KPI to derive the expected performance of Blink for any set of prefixes (Fig. 6). § B.3 provides more details about Blink's analysis using *TriScale*, which allows claiming with 95% confidence that, for at least 50% of the prefixes, Blink always detects link failures (TPR= 1) and reroutes traffic within 1 s (Fig. 9).

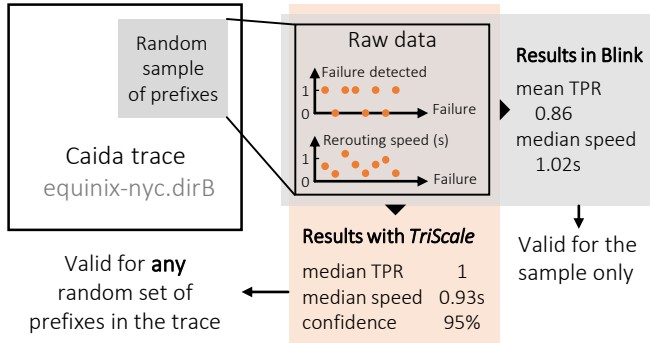

Figure 6: Using data from a sample of prefixes, *TriScale* allows generalizing and deriving performance estimates for any random set of samples from the same Caida trace [19]. *See Appendix § B.3 for further details.*

Takeaway 3. **Using *TriScale*, one can generalize performance results for a larger set of inputs.** *TriScale*'s methodology can handle any source of performance variability as long as the variability source can be reasonably modeled by a stationary distribution. Thus, one can use *TriScale* to generalize performance claims for evaluations based on network emulation: one can randomly select input traces or system parameters, and derive the expected performance of any other random set. However, the stationarity assumption cannot always be guaranteed (*e.g.*, for cross-traffic over the Internet), which is why *TriScale* includes an empirical independence test.

## 5.4  Video Streaming

This case study shows that the methodology of *TriScale* is easily compatible with common data reporting practices in networking, such as cumulative density functions (CDF).

In video streaming research, performance is often measured using the quality of experience (QoE) for the user as metric, for example, to compare state-of-the-art adaptive bitrate algorithms such as RobustMPC [88] or Pensieve [49]. Since QoE typically varies a lot, CDFs are often used to give a more global view on the performance of an algorithm. For example, Fig. 7 (area) shows the CDF achieved by Pensieve over a static set of synthetic network traces (reproduced from [49], see § B.4). However, CDFs are no different from other metrics: What is the confidence in the result? How much would it vary with a different set of traces?

A CDF is a representation of all percentiles of a given distribution. Hence, *TriScale* can be used to estimate an *entire CDF* by computing a large set of KPIs. For example, Fig. 7 (solid line) shows the 95% CI for the 2-th to the 98-th percentile, which provides a lower-bound on the expected performance. Hence, one can claim that, for *any* set of traces that would be generated/obtained similarly, the QoE of an algorithm is better than the CI CDF with 95% confidence.

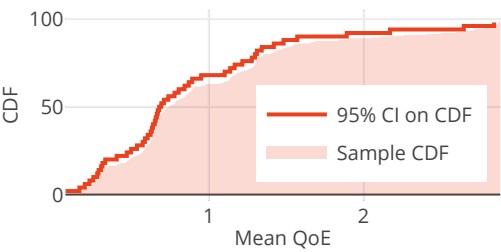

Figure 7: A CDF and its 95% CI, computed by *TriScale*. *Original CDF reproduced from [49]. The CI provides a lower-bound on the expected performance for any other random set of input traces generated similarly. Details in Appendix § B.4.* Online: tiny.cc/triscale-plots#Figure-7

Takeaway 4. **Percentiles are useful to evaluate any performance metric.** Using percentiles as KPIs makes *TriScale* metric-agnostic. Any variability source that can be modeled as a stationary distribution can be handled.

## 5.5  Scalability of *TriScale* Data Analysis

The data analysis proposed in *TriScale* induces no significant overhead. The computation time for the data analysis scales linearly with the input size, and it is fast (less than 1 s for one million data points on a commodity laptop): this will almost always be negligible compared to the data collection time. We evaluated the scalability of *TriScale* by measuring its computation time, *i.e.*, we measured how the time needed for the data analysis scales with a growing input size. To this end, we only considered the time required for performing computations, and exclude other outputs such as logs and plots (*e.g.*, Fig. 3a). See Appendix § C for more details.

## 6  Using *TriScale*

In this section, we provide concrete recommendations to select *TriScale*'s parameters (*e.g.*, confidence levels), as well as how to accommodate unusual metrics or experimental setups.

**Choosing percentiles and confidence levels.** In statistics, a confidence level of 95% is often considered standard. It is not recommended to use a lower value unless you are strongly limited in the number of runs you can perform. In any case, never go below 75%—at this level, an estimation has one chance out of four to be wrong!

For KPIs, use the median as a default. If you are interested in tail performance, consider more extreme percentiles (*e.g.*, 10th or 90th). If you want to show that some metric is small (*e.g.*, some delay), then use a large percentile (*e.g.*, 90th); the KPI is an upper-bound for that percentile, hence, you can conclude, *e.g.*, that 90% of runs are expected to have a delay at most as large as the KPI value. For variability scores, use the median. It is unlikely that you can perform so many series to be able to do anything more ambitious than that.

From a percentile and confidence level, *TriScale*'s API [36] returns the minimal number $N$ of runs (or series) that you must perform (Eq. (4)). Do more if you can, as more runs improve the reliability of the empirical independence test (§ 4.5). To increase robustness against outliers, *TriScale*'s API can also return the minimal $N$ with $r$ samples excluded from the CI.[8]

**Does *TriScale* support any metrics?** *TriScale*'s methodology is applicable to *any performance metric*. While its convergence test is not directly applicable to cumulative metrics

---

[8]There is no closed form expression like Eq. (4) for this calculus; thus, *TriScale* simply computes Eq. (2) with $m = r$ and increasing $N$ until the desired confidence level is reached.

(*e.g.*, energy consumption), this can be worked around by converting them to rates (*e.g.*, power draw; that is, the energy consumed over time) when testing for convergence.

In certain contexts, the *time to convergence* of a system is an important metric, *e.g.*, the recovery time after a link failure. The definition of "convergence" in such cases is context-dependent, and it is orthogonal to *TriScale*. One *could* use *TriScale*'s convergence test on an increasing slice of raw data, stop when the test passes, and use that as measure of the time to convergence. However, *TriScale*'s test is not designed for this purpose; it may not be the best, nor even a good idea.

**Defining runtime.** There are two default ways to define an appropriate runtime:

- If your experiment consist in fulfilling a given task, *e.g.*, transmit 1 GB of data, then the runtime must be long enough for the task to complete.
- If your experiment aims to estimate steady-state performance, *e.g.*, the power draw of a Wi-Fi network, then the runtime must be long enough for the metrics to converge.

One cannot a priori say how long a run should be; you must try it out and validate it. One possible approach is to perform a few (excessively) long runs and observe how long it takes for the task to complete, or for the metrics to converge; this provides a reasonable estimate for an appropriate runtime.

To decide whether you need convergence for your experiment, ask yourself the following question: Is performance expected to be the same if you increase the runtime by, say 10×? If yes, you should test for convergence.

Note that the runtime does not always need to be the same for all runs. For example, in video streaming, a run would be one video session, and different sessions have different durations. The runtime then simply becomes a factor that should be randomized and sampled in an *i.i.d.* way.

> **Ask for advice.** If the above recommendations are insufficient to nail down your experiment design, you can always ask for advice! An online forum is available for that purpose: https://groups.google.com/g/triscale

## 7 Discussion and Future Work

**Data collection.** *TriScale* is not responsible for the execution of networking experiments: it does not perform the data collection. Other frameworks such as Pantheon [87] or Puffer [86] are specialized in data collection; other examples include low-power wireless testbeds [47, 67, 68] and networking facilities [8, 26, 57]. *TriScale* can be integrated into these frameworks to create a fully-automated experimentation chain and build full-fledged benchmarking infrastructures, as envisioned by some networking communities [13]. In such a system, *TriScale* could be used as part of a feedback loop that would perform additional runs until a sufficiently narrow CI is obtained; *e.g.*, until a given replicability target is reached.

*TriScale* does not account for specific features of testbeds or data collection tools, such as those discussed *e.g.*, in [75]; this is intentional, as it would otherwise restrict the scope and applicability of the methodology. Moreover, one cannot magically "salvage" a poorly designed evaluation or an unstable experimental setup; what *TriScale* can do, however, is to observe and assess whether the chosen parameters and experimental setup eventually lead to replicable results. Furthermore, the links between the design and analysis phases of our methodology allow to rationally advise on how to design an experimental evaluation that is likely to produce replicable and trustworthy results—a unique feature of *TriScale*.

**Human-in-the-loop.** *TriScale* automates the data analysis and implements tests that verify whether the required hypotheses hold. However, it is up to the user to critically assess *why* tests fail when they do (*e.g.*, because the runtime should be longer—§ 5.1), and derive corresponding countermeasures (*e.g.*, pruning the start-up time in the raw data). Furthermore, some feedback and iterations are likely between the first set of tests and the final evaluation, as a larger set of experiments often uncover insights such as unknown correlation or seasonal components in the system being evaluated.

**Ranking solutions.** *TriScale* measures performance, but it does not rank. The evaluation results are always relative to a specific network or evaluation scenario (*e.g.*, a given cloud provider [75]). It is not trivial to claim that a solution A is generally better than a solution B. This problem relates to benchmarking and multi-objective optimization, which goes beyond the scope of *TriScale*.

**Community guidelines.** *TriScale* formalizes evaluation objectives (§ 4.4), but it does not dictate which parameters should be used. Similarly, *TriScale* quantifies the replicability of an experiment rather than concluding whether the evaluation is replicable (§ 4.7). Building on *TriScale*'s formalism, networking communities can now more easily set their own standards, metrics, reference parameters, and acceptable requirements in order to make performance evaluations more comparable, as it has been done in other disciplines [30].

**Other instances.** In this paper, we present a *methodology* to design and analyze performance evaluations. *TriScale* is one concrete *instance* of that methodology, composed of a well-chosen set of statistical approaches. We do not claim that *TriScale* is the only valid nor the best possible instance of the underlying methodology. In some context, the systems to evaluate may have a behavior benefiting from or requiring other models or statistics. Relevant examples include the definition of convergence (which may be different depending on the system), the normalization of measurements using different scaling function, the sampling of runs within series and series within sequels (periodic vs. random vs. biased random), as well as the availability of knowledge about the distributions or other statistical properties of measurements.

In such cases, one can build a different instance of *TriScale* based on the same methodology—*i.e.*, the concept of separating the variability sources into different time scales and addressing them independently—which can be generally applied with different statistical approaches. The specific instance we present in this paper does already appear well suited for a large class of performance evaluation scenarios, as exemplified by the case studies in § 5.

**Limitation.** In networking applications, many variability sources are time-dependent; this can be explained by the correlation (positive or negative) of a large portion of network traffic with human activity. This motivated the use of time scales to structure the *TriScale* framework. In some cases however, such time-based structure may not be the most appropriate; for example, for research on attack protection, or failure mitigation. Occurrences of such events are (a priori) not strongly time-dependent. This is also true for single events, such as the COVID pandemic and its impact on the Internet, or the 2021 Facebook outage [51]. For such studies, even the notion of replicability is debatable in itself.

Under the "Limitation" sub-section in Section 7, you state "The TriScale framework is designed with networking applications in mind, where the main variability factor is time—or can be modeled as such". I think this is a strong claim which is not well clarified. Network characteristics can vary based on link bandwidth, delay, and a variety of other factors. I believe TriScale can work in many such scenarios. The emphasis on time is not clear here. Security lapses are a nice counter-example. But is this the only category that cannot be handled by TriScale? It will be helpful if you can add more information on limitations.

However, even in such cases, the principles underlying *TriScale*'s methodology are still useful: with proper statistics, one can generalize performance claims based on a sample, as we illustrated with one of our case studies, related to failure detection (§ 5.3). In this example, the distribution we sample from is not time-dependent: it is a fixed distribution of prefixes in a set of traffic traces. There, KPIs allow generalizing performance observations to all prefixes, based on a sample.

In summary, while the full *TriScale* framework is not appropriate to all use cases, its core principles—*i.e.*, randomizing factors and using non-parametric CIs—are useful for any applications with performance variability, as long as we have some control over the factors to randomize.

**Multi-objective evaluation.** In this paper, we consider performance dimensions independently of each other. In many cases though, one is interested in comparing performance over multiple metrics (*e.g.*, delay and throughput). The concept of Pareto-dominance can be useful to extend the methodology to multi-objective performance evaluations. Moreover, networking metrics are often correlated (Fig. 1a), which should be accounted for in a multi-objective setting. How to address this is a complex question that we leave for future work.

In addition, one may be interested in two-sided intervals for KPIs, such that the expected performance is both upper- *and* lower-bounded. Currently, *TriScale* supports this by defining two independent KPIs; again, a better way to account for the correlation between these bounds would be interesting.

**Use what you need.** In this paper, we describe a full pipeline leading to a quantification of replicability. One may not always be interested to go all the way and assess replicability for every experiment; maybe you do not need convergence of runs, or maybe you do not need a strong statement on replicability. Each statistical method we present tackles a specific problem and is individually useful. Use what you need for your own experiments.

> **Do what you can.** Assessing replicability is hard: to make meaningful statements, one requires many runs, spread across large time periods, and often scheduled randomly. This might be impractical (if not impossible) to do on shared experimental platforms or testbeds. But that's what you need for replicability! If it is not possible, fine; but make sure not to overstate your performance claims!

## 8  Related Work

The replicability of experiments and comparability of results are cornerstones of the scientific method. In recent years, several studies have highlighted the inability of researchers from various disciplines to replicate their own experimental results [7, 60], often due to sloppy research protocols and faulty statistical analysis [12, 14, 66]. This is a problem in computer science as well [23, 76], where experiments are seldom replicable and artifacts rarely shared.

**Promoting replicability.** Recent work demonstrated that poor experimental and statistical practices has led to wrong or ambiguous conclusions. [75] presented a survey of recent cloud computing works and concludes that more than 60% of papers reports poor or no specification of the experiments, and that three-quarter of those that do are using fewer repetitions than necessary to mitigate the performance variability of cloud infrastructure. The Puffer project [86] showed that, for adaptive bit-rate algorithms, even with 2.5 years of data, the size of 95% CI of some performance metrics—*i.e.*, the uncertainty—is of the same scale as the performance "improvements" claimed in the original papers. To address this "replicability crisis" [7], many efforts aiming to incentivize a rigorous experimentation have gained momentum in computer science, including *e.g.*, ACM's badging system for publications [1]. In the networking community, especially challenged by the need to carry out experiments in dynamic and uncontrollable conditions [18, 53], several workshops [5, 16, 32], surveys [29], and guidelines [6, 44, 54, 65] have raised awareness on the replicability problem and promoted better experimentation practices. This large body of work mostly offers

*qualitative* statements on how an experiment should be performed and documented. Such statements emphasize, *e.g.*, the need to carefully choose when and how often to sample data [6], or suggest which methodology to adopt during performance evaluations [44]. However, there is no guarantee that following these recommendations leads to replicable results, nor a concrete way to assess whether an experiment can be considered replicable.

In contrast, *TriScale* provides researchers with *quantitative* answers about how to concretely design an experimental evaluation (*e.g.*, how many runs should be performed and how long they should be), which are derived from a clear experimental methodology grounded on robust non-parametric statistics. Moreover, *TriScale* offers a way to assess and compare the replicability of experimental results using clear performance indicators and variability scores.

**Supporting replicability.** Many experimental facilities and tools have been developed to aid researchers in carrying out replicable networking studies [57, 71]. Testbeds such as Emu-Lab [79] and FlexLab [63], as well as emulation tools such as MiniNet [33] and the mini-Internet [34], enable the creation of artificial network conditions using a given specification or passively-observed traffic. Emulated conditions offer a more controlled environment than experiments with real-world traffic (*e.g.*, by transmitting data over the Internet [11, 22], cloud [15, 26], or wireless interfaces [2, 31, 52]). However, even emulation suffers from performance variability caused by the underlying hardware and software components, which hampers replicability [50]. To overcome these problems, several solutions have been proposed [27], such as revisiting OS libraries [72], using virtualization [33, 42, 43], adaptable profiles [64], and fault patterns [4]. For "real-world" evaluations, other tools have been developed to support the replicability of mobility experiments [8, 20], interference generation [69], web pages load time estimation [77], live-streaming measurements [89], high-frequency wireless throughput estimations [3], and to enable researchers to consistently evaluate congestion-control schemes [87] or adaptive bit-rate (ABR) algorithms [86]. Note that real-world testbeds like Puffer [86] cannot control their test environment. Instead, they support replicability by randomly assigning ABR algorithms to video sessions and build statistical confidence out of massive amounts of data (over streaming 2000 h/day in the past two weeks[9]). We cannot hope to replicable this for every performance evaluation; *TriScale* offers a different approach.

The aforementioned tools aim to improve replicability *during* the experiments, while *TriScale* assists researchers *before* and *after* their execution. It does so by informing about the number and length of runs necessary to reach a given level of confidence, as well as by computing a score quantifying the variability of the results. Hence, *TriScale* complements the existing body of literature promoting and enhancing replicability in networking research. The most similar proposal to *TriScale* is CONFIRM [50], a tool aiming to indicate how many runs are required when running cloud experiments in order to obtain CIs of a given size; *e.g.*, ±1% of the empirical median. CONFIRM uses the same statistical approach to compute CIs as *TriScale* (see § 4.5) but it also requires extensive domain-specific knowledge about cloud environments in order to predict the expected width for the CIs. By contrast, *TriScale* is more general: it indicates, for any networking context, how many samples are required to compute a CI, but it does not say anything about the expected interval size, which can only be known a posteriori. Nevertheless, it is naturally possible to specialize the methodology (*e.g.*, different statistics, CI size targets, etc.) for specific contexts.

## 9 Conclusions

A consistent methodology for the design and analysis of experiments is crucial for a more rigorous and replicable scientific activity. In a prior workshop paper [39], we have argued that such a methodology is of paramount importance for networking, which is especially challenged by the need to carry out experiments in dynamic and uncontrollable conditions. *TriScale* is the concrete realization of our vision into a tangible framework: it implements a methodology grounded on non-parametric statistics into a framework that aids researchers in designing experiments and analyzing data. In addition, *TriScale* improves the interpretability of results and helps to quantify the replicability of experimental evaluations.

Beyond academic research, *TriScale*'s methodology may be beneficial, *e.g.*, for monitoring tasks performed daily by network operators or, more generally, for any performance evaluation run in a stochastic environment. We hope that *TriScale*'s open availability and usability [36, 38] will foster better experimentation practices in the short term and for the networking community at large. The quest towards fully-replicable networking experiments remains open, but we believe that *TriScale* represents an important stepping stone towards an accepted standard for networking experiments.

## Acknowledgement

It has been a long and bumpy road to get this work published; there were many people who helped us along the way and that we would like to thank. Hanspeter Schmid, for introducing us to non-parametric statistics, providing code snippets to implement the Thompson's CI computation, and fruitful stat discussions. The ETH statistical consulting team for their time and advice, and—most importantly—confirming that we were facing a non-trivial problem. Antonios Koskinas, Balz Maag, Ramona Marfievici, and Usman Raza for their help and brainstorming in the early stage of this project. Zimu Zhou, Benjamin Friedrich, Jennifer Rexford, Ankit Singla, Olivier

---

[9] https://puffer.stanford.edu/results/

Bonaventure, and Stefano Vissicchio for their feedback on earlier versions of this manuscript. And finally, the NSDI'2020, SIGCOMM'2020, SIGMETRICS'2021, and CCR reviewers for their helpful and encouraging comments despite rejecting the paper; in particular, we want to thank Damien Saucez and Luigi Iannone for the engaging discussion and encouragements that followed our submission to CCR.

This work was supported in part by the German Research Foundation (DFG) through the Emmy Noether project NextIoT (grant ZI 1635/2-1)

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

## A    Details on the Implementation

One obstacle to the adoption of non-parametric statistics is the lack of support in current scientific libraries; for example, the computation of CIs for percentiles, although present in textbooks, has no public implementation available.[10]

Therefore, to facilitate the use and adoption of the methodology we propose, we have implemented *TriScale* as a Python module including all necessary functions to apply our methodology. *TriScale*'s API contains one function for each timescale of the data analysis, with docstrings containing detailed information about each function's usage. Each function is essentially a wrapper that calls the statistical tests and methods appropriate for each timescale. The package also includes support tools such as data visualizations. *TriScale* uses Plotly [62] to create interactive plots in which one can zoom in and out, toggle the visibility of individual traces, read data values on hover, etc. Most plots in this paper have been produced using *TriScale* and all are "clickable": the figures are hyperlinks leading to dynamic versions of the plots. Our implementation is open source [36] and will be made available via common package management systems for Python. We use Binder [41] to provide an interactive demo of *TriScale* that runs directly in your web browser [38]—that's right, no need to install anything at all!

## B    Details on Case Studies

This appendix provides details on the four case studies presented in § 5; in particular, it details each evaluation scenario and how we have obtained the data. All case studies are performed using Jupyter notebooks, which are available in the *TriScale* repository [36].

### B.1    Congestion Control

**Reproducing the case study.** The entire case study is described in detail in a Jupyter notebook[11] that is available in the *TriScale* repository [36].

**Evaluation scenario.** This case study compares the performance of 17 congestion-control schemes using Pantheon [87]. We evaluate the throughput and one-way delay of full-throttle flows, *i.e.*, stable flows whose only throttling/limiting factor is the congestion control. For a fair comparison between the

schemes, we use the MahiMahi emulator [55] (integrated in Pantheon) and focus on a single flow scenario. We use only the calibrated path from AWS California to Mexico, provided by Pantheon.[12]

**Data collection.** We build the Pantheon toolchain from the source code provided by the authors[13] and test all schemes locally based on the aforementioned emulated network. We only modify the authors' code to save the throughput and delay raw data, such that we can do the analysis of runs using *TriScale*. We perform two sets of experiments with always 10 runs per series:

- A set of 5 series with a runtime of 30 s;
- A set of series with a runtime of 10, 20, 40, 50, and 60 s, respectively (one of each).

The data we collected are available on Zenodo [37].

**Results.** One important contribution of *TriScale* is to propose an approach to quantify the reproducibility of results with a variability score (§ 4.3). These scores allow making statements such as: *With 75% confidence, the variability scores give the magnitude of variation expected in the middle 50% of KPI values, shall one perform infinitely many series.*

The scores obtained for this case study are relatively small (less than 4 Mbps and 3 ms respectively; see Figure 8); this is expected as this case study is run in emulation, which provides replicable networking conditions for the performance evaluation. The entire case study contains more fined grained analysis, which is available in a Jupyter notebook[11] from the *TriScale* repository [36].

### B.2    Wireless Embedded Systems

**Reproducing the case study.** The entire case study is described in detail in a Jupyter notebook[14] that is available in the *TriScale* repository [36].

**Evaluation scenario.** We run a simple evaluation of Glossy [28], a low-power wireless protocol which includes as parameter the number of retransmissions of each packet, called *N*. We investigate the impact of two values of *N* on the reliability of Glossy, measured as the packet reception ratio (PRR). During one communication round, every node in the network initiates in turn a Glossy flood and all the other nodes log whether they successfully received the packet. This is repeated for $N = \{1, 2\}$. In addition:

- The evaluation runs on TelosB motes[15] (26 nodes);
- The motes use radio frequency channel 22 (2.46 GHz, which largely overlaps with Wi-Fi traffic);
- The payload size is set to 64 bytes.

---

[10]We are currently working to include this functionality into SciPy.
[11]casestudy_congestion-control.ipynb

[12]pantheon.stanford.edu/result/6539/
[13]github.com/StanfordSNR/pantheon
[14]casestudy_glossy.ipynb
[15]www.advanticsys.com/shop/mtmcm5000msp-p-14.html

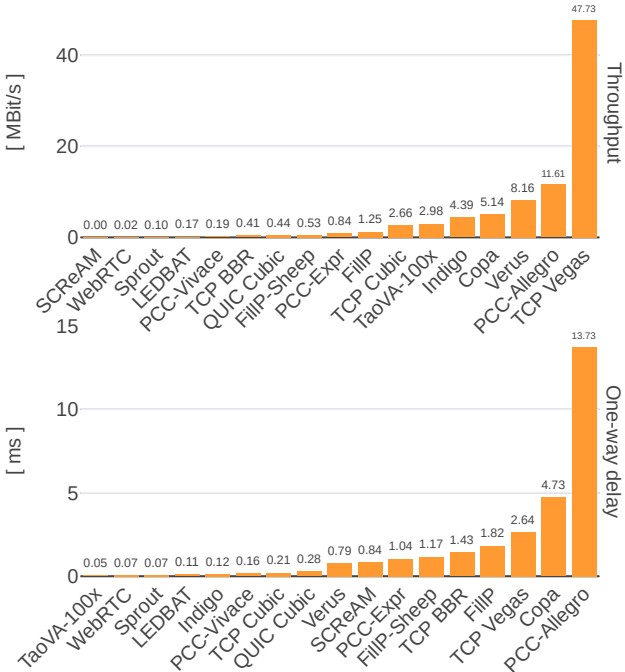

Figure 8: Variability scores computed by *TriScale* for the performance dimensions throughput and delay. *In this case study, the variability scores are computed as the 25th to 75th percentile interval estimated with 75% confidence. From the variability scores, the user gets a quantification, with a 75% probability, of the range of variation in the KPI values for 50% of the series. Hence, the variability scores quantify replicability: the larger the scores, the less replicable the results are.* Online: tiny.cc/triscale-plots#Figure-8

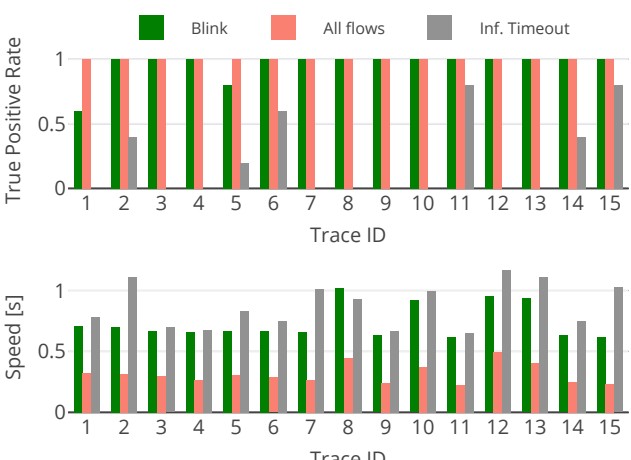

Figure 9: KPIs for Blink's performance evaluation. *95% CI on the median. Internet trace IDs listed in [35].*
Online: tiny.cc/triscale-plots#Figure-9

**Data collection.** We perform the experiments using the Flock-Lab testbed [47]. For both settings of the number of retransmissions *N*, we perform 24 randomly scheduled tests per day during 7 consecutive days. The data we collected are available on Zenodo [37].

### B.3 Failure Detection

**Reproducing the case study.** The entire case study is described in detail in a Jupyter notebook[16] that is available in the *TriScale* repository [36].

**Evaluation scenario.** This case study re-uses one of the evaluation scenarios from the original Blink paper (§ 6.1 in [35]). It considers 15 publicly available real Internet traces [19, 21]. For each trace, 30 prefixes are randomly selected among those that contain sufficiently many active flows. For each prefix, the characteristics of the traffic are extracted and used to run simulations where traffic sources generate flows exhibiting the same distribution of parameters than the one extracted from the real traces. Artificial failures are introduced in the simulation, which Blink tries to detect. Blink is compared against two baseline strategies:

- *All flows.* It monitors up to 10k flows for each prefix and reroutes if at least 32 of them sees retransmissions within the same time window. This strategy provides an upper-bound on Blink's ability to reroute upon actual failures, but ignores memory constraints.

- *Infinite Timeout.* It is a variant of Blink where flows are evicted when they terminate (with a FIN packet) and never because of the flow eviction timeout. This strategy tests the effectiveness of Blink's flow eviction policy.

**Data collection.** The authors of Blink kindly provided the data they collected for the original paper [35]. The data are now available on Zenodo [37].

**Evaluation objectives.** Each prefix is used to generate five failure scenarios, based on which we compute two metrics: (i) the true positive rate (TPR), *i.e.*, the ratio of failures that Blink successfully detects (out of 5); (ii) the median rerouting speed, *i.e.*, the time Blink takes to reroute traffic once it detects the failure. For both metrics, we use the 95% CI on the median as KPI, computed over the set of prefixes for each Internet trace.

**Results.** Blink achieves a TPR KPI of one for all the Internet traces, with a rerouting speed ranging between 0.5 to 1 s (Fig. 9). Hence, we can claim with 95% confidence that these are the minimal performance expected for Blink for any random set of prefixes within each of the Internet trace.

---

[16]casestudy_failure-detection.ipynb

## B.4  Video Streaming

**Reproducing the case study.** The entire case study is described in detail in a Jupyter notebook[17] that is available in the *TriScale* repository [36].

**Evaluation scenario.** This case study re-uses one of the evaluation scenarios from the original Pensieve paper (§ 5.2 in [49]). Specifically, it compares Pensieve against pre-existing adaptive bitrate algorithms using different quality of experience (QoE) metrics. The comparison is performed using the MahiMahi [55] network emulator by replaying a set of synthetic traces generated from real-world broadband datasets. We consider the set of traces generated from the FCC dataset;[18] these traces were created by the Pensieve authors by concatenating randomly-selected traces from the "web browsing" category in the August 2016 collection. There are multiple definitions of QoE: we consider the "linear" one (see [49] for details).

**Data collection.** The authors of Pensieve were not able to provide the data they collected for the original paper [49]. Consequently, we retrieved the QoE data directly from the paper plots using a web-based application.[19] The data we retrieved are available on Zenodo [37].

**Evaluation objectives.** From the QoE metric values, we compute the 95% CI (lower-bound) for the $\{2, 4, 6 \ldots 98\}$th percentiles, based on which we obtain a 95% CI for the entire CDF of QoE for the different algorithms.

**Results.** Fig. 10 shows the 95% CI CDFs computed for the linear QoE metric. The 95% CI are relatively close to the empirical CDFs, as illustrated in Fig. 7, which shows both the empirical CDF and its 95% CI for Pensieve (the same applies to all algorithms).

## C  Details on the Scalability Evaluation

This appendix provides additional information about the evaluation of *TriScale*'s scalability presented in § 5.5. We perform the evaluation using a Jupyter notebook[20] (*i.e.*, an open-source web-based interactive computational environment to create and share documents containing live code, equations, visualizations, and text) that is available in the *TriScale* repository [36]. Such evaluation, which we run on a commodity laptop, yields the results summarized in Table 3.

**Results – Metrics.** The data shows two modes in the execution time of the *analysis_metric()* function: a step increase, followed by a slow linear increase. This can be easily explained: the more computationally expensive part of

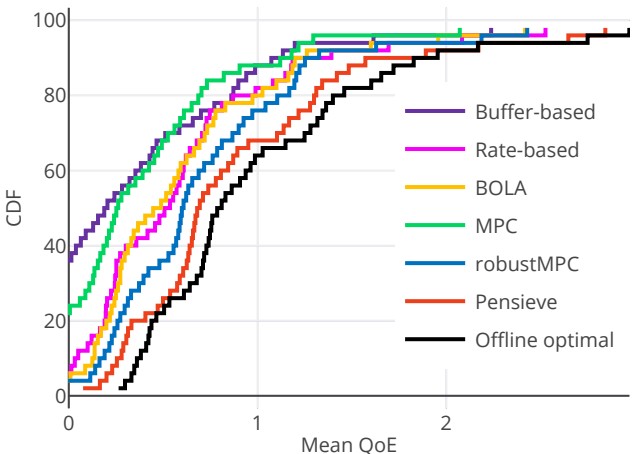

Figure 10: 95% CI on the CDF of various adaptive bitrate algorithms.                   Online: tiny.cc/triscale-plots#Figure-10

Table 3: Scalability evaluation. *TriScale data analysis is fast and scales well with increasing input sizes. The most time-consuming element is the convergence test (§ 4.5), which is performed before the computation of metrics. Still, it generally takes less than one second for inputs (i.e., the number of raw measurements in a run) of up to one million data points.*

| Computation of | Input size | Execution time (approx.) |
|---|---|---|
| Metrics | 1000 | 20 ms |
|  | 10 k | 50 ms |
|  | 1 M | 1 s |
| KPIs and Variability scores | 100 | 10 ms |
|  | 1000 | 100 ms |

---

[17] casestudy_video-streaming.ipynb

[18] Federal Communications Commission. https://www.fcc.gov/reports-research/reports/

[19] apps.automeris.io/wpd/

[20] triscale_scalability.ipynb

*analysis_metric*() is the convergence test, which includes the Theil-Sen regression (§ 4.5). The latter works by computing the slopes between all pairs of points and returns the median slope value; thus, the regression scales with $O(n^2)$.

However, *TriScale* does not perform the regression on the input data directly. Instead, *TriScale* divides the input data in chunks. For each chunk, a metric value is computed, leading to a new data series of metric values. The purpose of the convergence test is to verify that these metric values have converged; thus *TriScale* executes the Theil-Sen regression on this new data series. The Theil-Sen regression does not require many samples for producing a reliable result; a few tens of data points are often considered sufficient [82]. Thus, we can cap the size of metric data series (*TriScale* caps it to 100 values – § 4.1), which bounds the execution time of the Theil-Sen regression. Ultimately, this allows the *analysis_metric*() function to scale very well with the sample size.

The linear increase for a large number of raw samples is due to the computation of the metric on increasingly large chunks. The more complex the metric is, the longer the execution time. In this evaluation, a percentile is used as metric, which is computed efficiently with NumPy [56].

Overall, running *analysis_metric*() takes about 1 s for up to one million data points. The data collection time depends on the networking experiment, but it is unlikely that many experiments would produce much more than a million of data points per second. Thus, we conclude that the computation time of the *analysis_metric*() function is negligible for networking experiments.

**Results – KPIs.** The data shows a clear linear correlation between the sample size and the execution time of the *analysis_kpi*() function, which is not surprising: most computations are related to the determination of the confidence interval using Thompson's method, which is an iterative process through the ordered data samples [74].

The input size for the KPI computation is the number of series one performs for an experiment. Our results show that the computation takes less than 100 ms for an input size of 1k; we thus conclude that the computation time of the *analysis_kpi*() function is negligible for networking experiments.

**Results - Variability scores.** Unsurprisingly, the data is very similar as for *analysis_kpi*(): The two functions essentially perform the same computations. They only differ in the generation of outputs (logs and plots). Since the outputs are not considered in this scalability evaluation, we obtain very similar results for both functions. Thus, we conclude that the computation time of the *analysis_variability*() function is negligible for networking experiments.

