# OpenReview forum: "Designing Replicable Networking Experiments with TriScale"
_JSYS/2021/Aug_Papers — Accept_

### Official Review · Reviewer_GWWe · 2021-09-10
**A tool paper that facilitates replicable experiments and potentially benefits networking researchers**

**Decision:**

Weak accept: good paper with flaws that can be fixed in three months

**Review:**

### Pros
* Generally well-written and relatively easy to follow
* Useful, could potentially benefit all scientific researchers
* Methodology is grounded from strong theoretical analysis

### Cons
* Methodology is a bit high-level and lacks more detailed solutions
* More like a general experimental design tool instead of in the networking domain

### Detailed comments

Thank you for submitting TriScale to JSys. I enjoyed reading your paper learning how
foundational insights in statistics can be applied to networking experiment design. While networking researchers have been working hard to come up with new solutions and conduct extensive evaluations, it’s also the time for us to have a better understanding and more careful examination of experimental results, and that’s where TriScale could potentially help.

Meanwhile, I have a few comments below, which I hope could further improve the paper.

* It’s nice to break down experiment design into three questions and answer each one by one. However, I feel that the provided solution is a bit limited in what it can help. For example, given that convergence test does not always apply, oftentimes it’s still hard to know how long a run should be, e.g., for video streaming experiments what should be the total streaming duration in order to compare QoE for different streaming systems/algorithms? The paper uses long-term throughput as an example for congestion control evaluation, but sometimes we need to consider the impact of start-up time, say for short-lived flows/web pages. More importantly, a user testing a system oftentimes doesn't know whether she really needs the convergence test, and in this case it would be more helpful if the tool (TriScale) can give some hints. The authors mention that “Given the user’s objectives (e.g., the KPIs to analyze and the confidence levels to reach), TriScale helps answer questions...”, but it's still not very clear how to define the KPIs and choose the confidence level, e.g., should I choose mean, median, or some percentile? The paper also says “for each performance dimension, the user defines a metric together with its convergence requirements, a KPI, and a variability score”, but sometimes a user may not know how to best define them. It would be better if TriScale can give some suggestions on these low-level details, say according to how much computation resources/time a user has (e.g., lower down the confidence level or analyze only the median if one needs to get some quick results) or what the high-level goals of the experiments are.

* The methodology has high generality with little assumption of the underlying distribution. While this makes TriScale more widely applicable and easy to use in some cases, I still wonder if there could be cases where the distribution follows a specific model and TriScale could be tuned to gain improved analysis, especially when those cases are common in networking and the gain is worthwhile, e.g., more accurate claims with fewer runs. Also, if it’s emulation/simulation/testbed, how would the methodology be adjusted accordingly? Would domain-specific optimizations be beneficial, say web, video, congestion control, routing, wireless, etc? There may also be application specific requirements, e.g., in determining how long a run should be.

* Saying “Tools like Pantheon [84] support data collection, but leave the design of the experiments and the data analysis up to the researcher” makes the scope of “design of the experiments” seem limited to the three questions TriScale answers. However, it’s much more than that, e.g., including what network conditions and workloads to use. To me the paper is mostly on “when” (durations and specific start times) to run experiments instead of the entire “experiment design” which is a much bigger scope/problem to solve.

* In the congestion control motivating example, the authors mention that Copa data is not i.i.d., what’s the reason for that? And this seems to disobey “A common hypothesis is that the collected data is independent and identically distributed (i.i.d.).”

* Fig 1: The authors argue that “Contrary to what Fig. 1a suggests, we observe in Fig. 1b that TCP Vegas is not strictly better than TaoVA-100x”, however, Fig. 1a only shows that the mean throughput of Vegas is better than Tao’s, while Fig. 1b draws 25th throughput, so it’s more of a different metric than contrary. Also, it would be better to color the label of different schemes as well: some dots are too close to each other so it’s hard to tell which label/text corresponds to which dot.

* Need more details on how different works such as [84] handles uncontrollable environments. Some related ideas/tools for more controllable experiments are missing in the paper: Epload (https://www.usenix.org/system/files/conference/nsdi14/nsdi14-paper-wang_xiao_sophia.pdf) allows to control the variability of computation while also modeling page load dependencies for web browsing. Livelyzer (https://dl.acm.org/doi/10.1145/3458305.3463375) leverages virtual video capture to enable repeatable live video broadcast. Another study (https://ece.northeastern.edu/fac-ece/dkoutsonikolas/publications/pam21.pdf) uses a programmable 3-axis motion controller to collect traces in a controlled environment efficiently and in an automated way.

* “One should also consider robust statistics (e.g., using median instead of mean), i.e., statistics that are not overly skewed by outliers, which are common in experimental networking data.” would be good to have some examples on this.

* “Only users can know what is more appropriate for their evaluation, but it is important to understand this distinction when designing it.” again, the user may not always know, or at least prefer some tool that tells her such.

* Testbed classification table: Puffer [83] is more of a testbed for in-the-wild characterization/crowd-sourcing than addressing ​​uncontrollability of the experiments.

* analyzes “network condition data.” => analyzes “network condition data”.

* Does Triscale have applications beyond academic research? Say for network operators, application developers, or testing services?


**Expertise:**

Actively publishing in this area

**Useful:**

yes

---

### Official Review · Reviewer_UUNs · 2021-09-11

**Decision:**

Strong accept: excellent paper that will help the community

**Review:**

**Paper summary**

This paper proposes a methodology for the experiment design and data analysis of performance evaluations. In addition to test runs and series of runs that have been widely adopted, the authors introduce another time scale -- repetitions of series -- to capture the replicability of an experiment, via the definition of a quantifiable score ("variability score") with user-specified confidence level.

Furthermore, the paper instantiates the methodology as a software framework called TriScale and showcases the framework's usefulness with case studies.

**Strengths**
- First-of-its-kind paper to quantify the replicability of performance evaluations
- Developed a readily usable framework to implement the concrete methodology
- Great writing

**Weaknesses**
- TriScale's methodology has its own practicality issues (explained in the comments)

**Comments for author**

Thank you for submitting your work to JSys. I really enjoyed reading the paper. Its excellent writing clearly outlines the methodology as well as the statistical methods. I also appreciate the development effort of the tool TriScale.

TriScale is most useful in guiding networking researchers to design scientifically sound experiments, such as how long each run should be based on the convergence test, how many runs / series of runs to perform before meeting a specific confidence level, and the time span of a series of runs hinted by the network profiling. Therefore, TriScale could potentially help avoid design mistakes and also significantly foster reproducible studies.

*Issues and other comments:*
- Fig. 1a is indeed prone to misinterpretation, as the paper also incorrectly interprets one-sigma ellipses as "one can expect about 68% of the data points to fall in that region". The correct percentage is supposed to be 40% because each ellipse represents the confidence region of the sum of squares of two Gaussian-distributed variables, which follows a chi-squared distribution. The projections of such an ellipse on either dimension are indeed 68% confidence intervals, but the two-dimensional confidence region is about 40%.

- The KPI of common interest is often multi-dimensional, e.g., people want to know both the lower and upper bounds of a performance metric, or both the average performance and performance variation (robustness). It is unclear how useful the one-dimensional KPI of TriScale would be in practice. If TriScale extends the current KPI, e.g., if Fig. 1(b) reports two-sided CIs instead, then it will have the same issue as Fig. 1(a) when two schemes' CIs are overlapping, losing the claimed advantage "unambiguously compare different schemes". The issue is not easily fixable also because the variability score cannot seem to accommodate multi-dimensional KPIs. What would the score be if the KPI is a two-sided CI? This seems different from a multi-objective evaluation and has not been discussed in the paper.

- In the same vein, TriScale does not capture the correlation of metrics. Take Fig. 1 as an example again: The one-sigma ellipses in (a) suggest the correlation between throughput and latency (which are usually positively correlated), but the information is lost in (b). Again, Pareto-dominance does not address the correlation of metrics.

- TriScale can check whether a runtime passes the convergence test, but it does not really give a practical way for researchers to choose the runtime during the experiment design. For instance, the ramp-up time of LEDBAT's throughput would also depend on the bandwidth -- Does it mean that for each network profile, researchers have to increase the runtime repeatedly until the convergence test is passed?

- Despite TriScale's usefulness, the definition of "variability score" essentially repeats what researchers have been applying to a series of runs, treating each experiment as a run. The quantifiable replicability is not novel in this sense (maybe it does not need to be novel?).

**Expertise:**

Published in this area in the last 5 years

**Useful:**

yes

---

### Official Review · Reviewer_8WTe · 2021-09-13
**TriScale Review**

**Decision:**

Weak accept: good paper with flaws that can be fixed in three months

**Review:**


Summary:

The paper presents TriScale, a methodology for analyzing the replicability of experiments. The paper achieves this by partitioning the data into multiple timescales---runs, series, and sequels---and analyzing temporal characteristics of variability at these timescales. TriScale relies on “variability score,” an estimate of how similar the results are during a replication effort, to quantify the replicability. Four case studies of various network applications using TriScale is also presented.

Strengths:
+ TriScale tackles an important problem related to replicability of experiments in research
+ The systematic methodology presented in TriScale can improve the community’s confidence about research results presented in a paper and its replication efforts
+ The paper is well-written and easy-to-follow

Weaknesses:
- A more detailed discussion on metrics/settings that cannot be evaluated using TriScale will be helpful

Detailed Comments:

Thanks for submitting your paper to JSYS 2021.

TriScale takes an important step towards improving reproducibility efforts in the community. It can help researchers determine the number of runs, duration of runs, etc. required to ascertain the confidence in conclusions derived from an experiment. The paper is well-written and easy to follow.

There are a few aspects that can be improved in the paper.
* It is understandable that TriScale cannot determine the parameters that should be used for experiments. However, it will be useful to identify the categories of metrics that cannot be used with TriScale. For example, the paper mentions that the convergence test does not hold while measuring energy consumption since it is cumulative over time, and hence, one should measure power draw instead. Cumulative metrics cannot be tested for convergence. Are there other categories of metrics that do not fit under TriScale framework? It will be useful to identify them explicitly (also, highlight in the discussion section that cumulative metrics cannot be measured)

* Can TriScale be used with metrics that need a distributed view? For example, maximum queue length encountered by a packet on its path within a data center.

* TriScale employs specific techniques for each step (Theil-Sen linear regression for convergence test, Thompson’s method for confidence interval estimations, etc.). Are there alternatives? If so, is it easy to plug in another user-preferred function in place of the TriScale default functions?

* Network profiling for identifying seasonal components is interesting. But what if the seasonal components in the replication environment do not match with the seasonal components in the original environment. For example, the original experiment is cellular analysis performed in a business district and the replication effort is in a residential area. Can network profiling be used to raise an alarm that the replication environment does not match the original environment and that is the cause for a bad variability score?

* A category of experiments in networking involves testing convergence of protocols when users arrive/leave, links fail, etc. The time taken by congestion control (for example) to converge under these conditions is an important performance criterion. How will TriScale handle such metrics?

Overall, TriScale is a useful effort that can improve reproducibility efforts in the community. However, it will be useful to clearly describe the shortcomings of the system, particularly metrics that cannot be handled.




**Expertise:**

Follow the literature closely, last published 5+ years ago

**Useful:**

yes

---

### Official Review · Reviewer_Rpcn · 2021-09-14
**TriScale Review**

**Decision:**

Strong accept: excellent paper that will help the community

**Review:**

Thank you for submitting your work to JSys. The paper tackles an important and often overlooked problem of replicability in networking research. The effort is commendable, thorough, and can be used by other researchers as a template to effectively design experiments and report their results.

Strengths:
+ Significance: the paper addresses an important problem
+ Clarity: the proposed methodology is intuitive and easy to follow


Weaknesses:
+ Generalizability: It is unclear where exactly the proposed methodology will fail and why? Section 6 is not really helpful in understanding these limitations.

This paper addresses a critical problem in networking research, replicability. We all know many research artifacts published at top-tier conferences are hard to replicate. Designing experiments that report the performance of the proposed research artifacts with quantifiable confidence amidst inherent variability is non-trivial. Researchers often opt for simpler rules-of-thumb that were applied by previous works. These rules don't generalize, and applying them leads to hard-to-replicate experiments/results. More specifically, the lack of concrete methodology to determine the run/series duration and the number of sequences make it difficult to quantify the true performance of the proposed research artifacts. The methodology proposed in this paper will aid the researchers in evaluating existing and future research artifacts, promoting replicability.

In Section 6, it will help if the paper can expand on instances where the proposed methodology is not suited to evaluate the replicability of the proposed solution. I struggled to grasp the generality of the proposed methodology. Though I am glad to see the paper applying the proposed methodology to five different problems. It will be great to elaborate on what attributes of these problems enable applying the proposed methodology. Further, elaborate, with examples, on which type of problems (artifacts) will not benefit from Triscale.

**Expertise:**

Follow the literature closely, last published 5+ years ago

**Useful:**

yes

---

### Author Response · Authors · 2021-09-21
**Revision plan**

Dear reviewers,

Thank you again for your thorough reviews and constructive comments to help improve the work further. Here is a summary of the main points we intend to address in the final version.

- Several suggestions were made to give more concrete examples. We will address this by discussing, possibly in a dedicated section, how to, e.g.,
  - Choose meaningful KPIs and variability scores,
  - Define a strategy to select an appropriate runtime,
  - Handle unusual metrics such as time to convergence,
  - Decide whether testing for convergence is useful or not.
- Several of you mentioned that the limitations of TriScale (when it fails, what cases it is unable to handle) should be clarified. We will extend section 6 in that direction.
- A discussion on using alternative statistical methods within TriScale is already present (Sec. 6, Other instances) but it is fair to say that it could be extended a bit and refered to in other relevant places in the paper (typically, at the end of Sec 4.5).
- We will correct the wording of the introduction to break the (wrong) impression that "exp. design == defining metrics/KPI/var. scores"
- We will extend the comparison with Puffer in the related works, and will consider the additional references that were suggested.
- You wonder whether the TriScale's methodology could be specialized/optimized for specific use cases/netorking context. It is most certainly the case; one example of such specialization is CONFIRM [51] (already discussed in the related works) in the context of cloud computing. In contrast, TriScale is designed to be a general methodology applicable, as is, to a wide number to contexts. Hence, discussing specialization of the methodology seems a bit out of scope for this (already fairly long) paper.
- There is certainly a use for TriScale beyond academic research: any performance evaluation run in a stochastic/variable environment can benefit from a mrore sound statistical analysis. I could imagine that the monitoring tasks of network operators could fall in that category. However, this feels (to me) too hypothetical to make a strong case. It could be mentioned as a perspective in the discussion though.

Finally, there is the complex question of multi-objective evaluations and the correlation of metrics. It is very true that most evaluations are, in practice, multi-dimensional. As mentioned in the discussion (Sec.6 Multi-objective evaluation.), the current paper does not address this problem and considers each dimension separately. Although the paper may currently suggest it, using Pareto-dominance is not sufficient to trivially address this problem, as you pointed out. We have done some further research regarding this problem, but decided to leave it out of the TriScale work, as (i) it is a related but somewhat independent problem, (ii) TriScale alone is already a mouthful of content.

Naturally, we will also address the other error-corrections and enhancements that you pointed out in your reviews.
If you believe there is there is anything else that would further improve the paper, please feel to mention it. And of course, we are open for further discussion regarding any of these points!

---

> ### Comment · Reviewer_8WTe · 2021-09-23
> **Shepherd's response**
>
> Thanks for the detailed response. The proposed changes look good to me. Please highlight the diff when you send the paper. Looking forward to the updated version. Thanks!

---

> ### Author Response · Authors · 2021-10-14
> **Revised version now available**
>
> Dear reviewers,
>
> Thank you for your patience. I just uploaded the revised version of the paper, addressing your comments as discussed above, including an highlighting of the main modifications in the text. The current page layout is imperfect. The diff highlighting I used creates some weird issues, and since the layout will change after we put the author' names in, I did not bother to fix it just yet.
>
> Please let us know if you have any further comments you would like to see addressed. Thanks again for your help in making the paper better!

---

> > ### Comment · Reviewer_8WTe · 2021-10-15
> > **Response from Shepherd**
> >
> > Thanks for submitting the updated version. I have a few comments.
> >
> > - Under the "Limitation" sub-section in Section 7, you state "The TriScale framework is designed with networking applications in mind, where the main variability factor is time—or can be modeled as such". I think this is a strong claim which is not well clarified. Network characteristics can vary based on link bandwidth, delay, and a variety of other factors. I believe TriScale can work in many such scenarios. The emphasis on time is not clear here. Security lapses are a nice counter-example. But is this the only category that cannot be handled by TriScale? It will be helpful if you can add more information on limitations.
> >
> > - Reviewer GWWe had pointed out several related papers on the controllability of experimental environments. Please add a discussion on that.
> >
> > - The sub-heading "Using unusual metrics" is a bit vague. Maybe change to "Can TriScale handle all metrics?" The description could be reworded too: "Even if the system should converge, it is no limitation to the use of cumulative metrics, such as energy consumption or packet counts; these metrics can be trivially converted into rates (e.g., power draw; that is, energy consumed over time), which should converge."
> > Maybe "While handling cumulative metrics, we recommend converting them to rates (e.g., power draw; that is, energy consumed over time) for measuring convergence" or something along those lines?

---

> > > ### Author Response · Authors · 2021-10-15
> > > **Quick answer**
> > >
> > > Thank you for the quick feedback.
> > >
> > > - That's a fair point, the statement is too strong given the (lack of) arguments given to support it, I'll adjust that. It's actually quite hard to think about what a tool _cannot_ do when you tried your best to design it to be general... Security is probably not the only case, I'll make that clearer.
> > > - I included the suggested references in the "supporting replicability" paragraph of the related work. There are indeed relevant, but in my opinion not so related that they deserve a longer discussion (like we do for CONFIRM). Many similar references are mentioned; in my opinion, this paper is not meant to discuss them all. Do you agree?
> > > - Fair enough. I'll integrate the proposal.
> > >
> > > I do not have time to work on these update right now; I will post an updated version at the latest later tonight.

---

> > > ### Author Response · Authors · 2021-10-15
> > > **Revised version submitted**
> > >
> > > I just uploaded a new version with the suggested revisions to the "unusual metrics" and "limitations" paragraphs. Please let me know in case you have any further comments you would like to see addressed.

---

> > > > ### Comment · Reviewer_8WTe · 2021-10-18
> > > > **LGTM**
> > > >
> > > > Thanks! This looks good to me.

---

### Meta-Review · Area_Chair_7hwn · 2021-09-15

**Recommendation:** Accept
**Confidence:** 4

**Metareview:**

The reviewers are overwhelmingly positive about the submission. There are minor issues pointed out in the detailed comments, and we believe the issues can be resolved in a short timeframe.

[Update on 9/17]
The reviewers agree that the paper sheds light on a crucial issue and appreciate the effort in building a concrete tool which is an important step in the right direction.

That said, there are plenty of concerns detailed in review comments, all of which require clarifications in the final version. Given that the deadline for "camera ready" is within a month, please share a plan on how you would address these comments soon and start working on addressing the comments immediately.

In the meantime, please do contact us (through comments here) if you have any question regarding the comments or the shepherding process.

---

> ### Author Response · Authors · 2021-09-20
> **Thank you all for your constructive feedback!**
>
> Dear reviewers,
>
> Thank you for all your helpful comments. We are very happy that you found our work interesting and potentially useful for the community.
>
> We will carefully consider your comments and lay-out our plan to address them promptly. I hope to get the all modifications done by the end of the week.

---

### Decision · Program_Chairs · 2021-09-16

**Decision:**

Accept

**Comment:**

Congratulations!